# Died with or Died of? Development and Testing of a SARS CoV-2 Significance Score to Assess the Role of COVID-19 in the Deaths of Affected Patients

**DOI:** 10.3390/diagnostics11020190

**Published:** 2021-01-28

**Authors:** Arianna Giorgetti, Vasco Orazietti, Francesco Paolo Busardò, Filippo Pirani, Raffaele Giorgetti

**Affiliations:** 1Department of Medical and Surgical Sciences, Unit of Legal Medicine, University of Bologna, Via Irnerio 49, 40126 Bologna, Italy; ari.giorgetti@gmail.com; 2Department of Excellence of Biomedical Sciences and Public Health, University “Politecnica delle Marche” of Ancona, Via Conca 71, 60126 Ancona, Italy; vasco.orazietti@gmail.com (V.O.); f.busardo@staff.univpm.it (F.P.B.); r.giorgetti@staff.univpm.it (R.G.)

**Keywords:** COVID-19 Significance Score (CSS), COVID-19, SARS CoV-2 related fatalities, death, cause of death

## Abstract

Since December 2019, a new form of coronavirus, SARS-CoV-2, has spread from China to the whole word, raising concerns regarding Coronavirus Disease 2019 (COVID-19) endangering public health and life. Over 1.5 million deaths related with COVID-19 have been recorded worldwide, with wide variations among countries affected by the pandemic and continuously growing numbers. The aim of this paper was to provide an overview of the literature cases of deaths involving COVID-19 and to evaluate the application of the COVID-19 Significance Score (CSS) in the classification of SARS CoV-2-related fatalities, comparing it with the Hamburg rating scale. The results obtained allowed us to highlight that CSS used after a complete accurate post-mortem examination, coupled to the retrieval of in vivo data, post-mortem radiology, histology and toxicology, as well as to additional required analyses (e.g., electronic microscopy) is a useful and concise tool in the assessment of the cause of death and the role played by this virus. A shared use of this scale might hopefully lower the inhomogeneities in forensic evaluation of SARS CoV-2-related fatalities.

## 1. Introduction

In December 2019, starting from the urban area of Wuhan, a new form of coronavirus, SARS CoV-2, began to spread firstly to a national level, and rapidly to the whole world. Its diffusion was so fast that on March 11, the World Health Organization (WHO) declared Coronavirus Disease 2019 (COVID-19) a pandemic [1]. At the time of the present article (December 2020), the cases of COVID-19 registered in the world have reached 68,679,195, with 47,583,441 patients completely recovered. Active cases amount to 19,530,028, 0.5% of which are in severe or critical conditions. Most affected countries include the United States, India and Brazil. Deaths related to COVID-19 amount to 1,565,726, with wide variations among countries affected by the pandemic and continuously growing numbers [2].

Global research efforts have been, since then, focused on studying the natural history of the disease, the immune responses, rapid and reliable diagnostic testing and on understanding the mechanisms underlying the clinical picture, with the aim of better treating affected patients and developing an effective vaccine as soon as possible. Although several vaccines have shown promising results in phases 2 and 3 of the experimental studies [3,4], long-term effects of COVID-19 and of related counteracting drugs and therapies might continue for years. The COVID-19 death toll is reported everyday nation by nation and is one of the main aspects of health surveillance, which guides health and social policies.

In this context, the distinction between “died from” and “died with” COVID-19 still represents an under-addressed and unsolved issue [5,6]. Often it is a difficult task for a medical practitioner to establish into which of these categories a death falls; indeed, distinguishing between “dying with” and “dying from” COVID-19 requires more complex investigation into the cause of a death, beyond citing a positive SARS-CoV-2 test. Although some clinical conditions, as well as laboratory and imaging alterations, are known to be associated with a worse outcome [7,8,9,10,11,12,13], there are still difficulties in classifying COVID-19 related deaths [14], due to the lack of consensus criteria. Throwing light on what is being counted as a COVID-19 death is also essential to understand the impact of the virus and to inform the public. According to the “International guidelines for certification and classification of COVID-19 as cause of death”, published last April by the World Health organization (WHO) and based on the International Classification of Diseases (ICD), a clinical-based categorization of COVID-19 deaths can be performed by recording a pathophysiological sequence of the clinical conditions leading to death, as well as other contributing causes [15,16]. Medical certifications of death compiled for research and surveillance purposes by treating physicians have relevant consequences, e.g., they might bias scientific studies for the development of clinical risk prediction models or prevent the development of public health safety measures [17,18]. COVID-19 might be a direct cause, an underlying cause of death or a contributing condition. Even probable infections are acceptable in death certificates and do not necessarily point to the need for a judicial autopsy or a coroner intervention, even if it is a notifiable disease [15]. Though most autopsies are not necessary for laboratory-confirmed deaths, [15] in the absence of a probable cause of death or when there is a suspect of medical liability, a post-mortem examination might be necessary [15,17]. It is well documented that the clinical cause of death might not coincide with the pathological one [19,20] and, notwithstanding increasingly accurate laboratory and instrumental techniques, the role of the autopsy still remains relevant [21]. Since this virus belongs to Hazard group 3, in the early stages, very few autopsies were performed on COVID-19 patients, leading to a loss of valuable information. With the aim of reducing biological risk for contagion, several guidelines have been developed [22,23,24,25,26,27,28,29,30,31]. While some authors have proposed the use of “special autopsy facilities”, others promoted a shift towards minimally invasive autopsies, performed by ultrasound-guided biopsies in different organs [32,33]. However, the latter may not provide a complete picture, making it difficult to answer questions about exact causes of death and SARS CoV-2 liability.

## 2. COVID-19 Significance Score (CSS)

As recently reported, a COVID-19 Significance Score (CSS) has been proposed [34]. The CSS classifies fatalities involving COVID-19 into four categories (as also suggested in other forensic disciplines, e.g., toxicology) [35,36]:0: COVID-19 is merely an occasion; it has no role in the patient’s death.1: A role of COVID-19 in the patient’s death cannot be excluded, although an alternative cause of death is likely.2: COVID-19 likely contributed to the death, together with other factors that may have played a prominent role.3: COVID-19 is the leading cause of death.U (unclassified or unclear), when not enough data are available, when further instrumental and laboratory tests are needed to clarify the situation or when the role of COVID-19 remains unclear despite all tests and analyses.

In the application of the CSS, the following features must be taken into consideration:Presence and severity of COVID-19, considering both in vivo and postmortem data (natural history of disease, results of upper and lower airway swabs, clinical records, laboratory tests)Presence and severity of comorbidities. It has been widely demonstrated that the presence of comorbidities is more frequently related to a different natural history in SARS-Cov-2 infection.Circumstances of death. External traumatic events, e.g., a fatal car accident, involving a patient infected by SARS CoV-2 might rule out the responsibility of the virus in the death. This might be less evident in suicides. Indeed, cases of Corona Suicide have been reported worldwide [37] and COVID-19 might play an indirect role, by ingenerating fear, burden and a sense of responsibility for having infected other people, especially the closest relations.Post-mortem imaging. Together with the tests performed in vivo or individually, post mortem radiology, including X-rays and post-mortem computed tomography (PMCT), may offer prominent information about the severity of the infection, as well as on any other alterations not clinically appreciable. However, due to the biological risk, these examinations should be carried out according to appropriate safety protocols and in any case should not hinder the hospital routine.Macroscopic and microscopic autopsy findings. Autopsy plays a central role in the development of this score, as information that can be obtained from this examination cannot be provided by any other imaging or laboratory test.Toxicological evaluation. A screening of the most common substances of abuse could be useful to exclude acute intoxications. As often happens in comorbid patients, the consumption of multiple drugs might lead to adverse events and exitus even at concentrations lower than the toxic levels, due to synergic effects. Insufficient dosages of a necessary drug might as well explain a death and these possibilities require a quantitative analysis.Additional analyzes. When required by the case in question, such as bacterial culture, virological tests and other specific exams.

The aim of the present work is to provide an overview of the literature cases of deaths involving COVID-19 and to evaluate the application of the CSS in the classification of SARS CoV-2-related fatalities, comparing it with other rating scales currently available

## 3. Material and Methods

### 3.1. Literature Review and Data Extraction

A systematic review of the available literature was performed, seeking information regarding post-mortem examinations performed on the deceased who had tested positive for SARS Cov-2, with the infection confirmed in vivo and/or after death (topic of the search and main inclusion criteria). Data was collected from May 2020 to August 2020, by performing a search on an international database (Pubmed) using the following search terms. The keyword “covid” (variably written) was linked through the Boolean operator “AND” to the following terms, alternatively: “autopsy”, “full autopsy”, “post-mortem”.

Inclusion criteria were: relevance to the topic; English, Spanish, German or Italian language; date of publication (up to 31 July 2020); retrievability of a full-text.

Papers that did not report the results of a post-mortem examination, e.g., publications dealing with the health care personnel safety or technical aspects of post-mortem examinations, cases analyzed only by post-mortem biopsies performed on a single organ and papers from which individual patient data could not be extracted, were excluded from the work (exclusion criteria).

A database with the results was created in Microsoft Excel and the following data, organized in two sections, were extracted.In vivo data: in addition to the patient’s personal details (age and sex), the history of the disease, any comorbidities, medications taken before and during the SARS CoV-2 infection, information about the swab, laboratory and imaging (e.g., X-rays and computed tomography or CT) data.Post-mortem data: death circumstances, any post-mortem imaging examinations, type of postmortem examination (full, partial, histology), macro and microscopic features emerged from the autopsy and related analyses, cause of death (when specified) and the role played by SARS CoV-2 as reported by the authors.

### 3.2. CSS Guiding Tool Development and Score Application

A short and easy guiding tool has been developed in order to facilitate the application of the CSS across original points 1 to 6 [34]. The features considered in this guide refer to the most frequent pictures described in literature to date. For this reason, the CSS is not to be considered as a definitive tool, but susceptible to modifications and improvements along with the progress of the state of the art.
in vivo and/or post mortem positivity of the swab for SARS CoV2 (YES/NO)
any reported symptoms compatible with COVID-19 (YES/NO) and severity of them (MILD, MODERATE = the situation required non-invasive techniques, SEVERE = the situation required invasive techniques, such as intubation)any symptoms referable to other causes (YES/NO)drug therapy administered during COVID-19any changes in laboratory tests compatible with COVID-19 (YES/NO)any alterations in the laboratory tests due to other causes (YES/NO)evidence of bacterial, fungal or viral superinfection (YES/NO)radiological evidence (X-ray, CT compatible with COVID-19) (YES/NO)presence (YES/NO) and number of comorbidities (1, 2 OR MORE)
severity of comorbiditiesdrugs consumed prior to COVID-19external traumatic cause of death (e.g., car accident, gunshot, electrocution, drowning) or suicidal/homicidal manner of death (YES/NO)*radiological evidence (XR, CT compatible with COVID-19) (YES/NO)
presence of any other pathological alterations (YES/NO)macroscopic and/or microscopic findings compatible with COVID-19 (YES/NO)
presence (YES/NO), type (as chronic obstructive pulmonary disease (COPD), cardiomyopathy, thromboembolism, coronary thrombosis] and severity of other pathological conditionspresence of drugs/substances of abuse (YES/NO)
presence and concentration of drugs taken regularly in chronic or during COVID-19.

* in case of suicide, the liability, even if indirect, for SARS CoV-2 cannot be excluded. Any useful information, such as a history of psychiatric pathologies, should therefore be evaluated carefully.

All literature cases were classified according to the CSS by three independent blinded investigators and the inter–rater agreement was assessed by non-parametric ANOVA.

### 3.3. The Hamburg Score

In a recent study performed at the University of Hamburg–Eppendorf, Edler et al. also proposed a classification system for deaths involving COVID-19 [38]. In the paper, the first 80 consecutive autopsies carried out on patients positive to COVID-19, who died in Hamburg, were reported. In fact, the approach used in the federal state of Hamburg is to examine all the deceased citizens with a confirmed SARS-CoV-2 infection, subjecting the bodies to a PMCT and performing a complete autopsy (by opening the three cavities and dissection of all organs). The results of the exams are then progressively uploaded to a national register, with the purpose of collecting data from all the autopsies performed in Germany on COVID-19 patients. Based on clinical information, PMCT and autopsy findings, the researchers propose a categorization of COVID-19 positive deaths, in order to determine whether the virus was the cause of death or whether exitus occurred independently from it.

This scale, just like the CSS, ranks COVID-19 related death into 4 categories: 1—definite COVID-19 death: autoptic pneumonia and/or acute respiratory distress syndrome (ARDS); 2—probable COVID-19 death: Autoptic pneumonia and/or ARDS and other infectious causes of death (e.g., pulmonary embolism); 3—possible COVID-19 death: cause of death that cannot be determined with certainty by autopsy (e.g., cardiac arrhythmia in cardiomyopathy) OR autoptic respiratory tract infection/pneumonia of other genesis (e.g., aspiration pneumonia, exacerbated COPD); 4—SARS-CoV-2 detection with cause of death not associated to COVID-19: Clear non-SARS-CoV-2-related cause of death (e.g., brain mass hemorrhage in hypertension, acute myocardial infarction in coronary thrombosis.

A death corresponding to categories 1–3 is defined as “COVID-19 death” (corresponding to CSS 3-1), therefore, COVID-19 related, while category 4 contains deaths not related to COVID-19 (corresponding to CSS = 0).

Within the post-mortem cases reported by the University of Hamburg–Eppendorf, the mean among the CSS assigned by the three raters and the Hamburg scores were compared by means of non-parametric *t*-test.

## 4. Results

### 4.1. Literature Review

Thirty articles were included in the present literature review, corresponding to 84 post mortem examinations. Results of the selection process are shown in Figure 1, and detailed data for each case are reported in Table 1. Fourteen studies describe multiple cases [39,40,41,42,43,44,45,46,47,48,49,50,51,52], while 16 were case reports [53,54,55,56,57,58,59,60,61,62,63,64,65,66,67,68]. The highest sample size corresponded to 14 cases.

Victims were mostly male (58 cases), with a mean age of 65.3 years (median: 70.5, lower limit: 17, 25% percentile: 55.5; 75% percentile: 76.0; upper limit: 91). Since the exact age was not reported in two cases, defined as “middle aged”, in the mean calculation they were both considered to be 55 years old. All the cases tested positive for SARS CoV-2 RNA, with swabs performed in vivo and/or post-mortem. Seventy-four victims had comorbidities, though no information was reported in 5 cases, and of these, the majority (56 cases) had 2 or more diseases, up to a maximum of 14. The most frequently reported were: arterial hypertension (in 41 cases, 55%), diabetes mellitus (in 28 cases, 37.8%) and obesity (in 24 cases, 32.4%).

Signs and symptoms of SARS Cov-2 infection were reported in 65 out of 84 cases. Respiratory symptoms were described in 50 cases, fever in 47, cough in 34. Fatigue/myalgia (11 times), gastrointestinal symptoms (10 times), alterations of consciousness, e.g., delirium/confusion/syncope (8 times), chills (6 times), tachycardia (5 times) and headache (4 times) were less common. Even more rarely hypotension, incontinence, chest pain and balance disorders (like dizziness/postural instability), described each in 2 cases, were reported. Acute kidney injury, acute cardiomyopathy, bradycardia, atrioventricular block, cardiac arrest, hemoptysis, hematuria, sore throat, sinusitis and anorexia were reported only one 1 time. In one case, unspecified systemic symptoms were described. Among the 49 cases whose laboratory abnormalities were reported, the most frequently encountered changes were elevated c-reactive protein (CRP) (51%), lymphocytopenia (46.9%), elevated lactate dehydrogenase (LDH) (38.7%), elevated creatinine (26.5%), elevated D-dimer (24.4%), leukocytosis (22.4%). Other alterations, such as in the values of fibrinogen, troponins, ferritin and Il-6, were rarely found.

Data on medications were available in 13 patients, 12 of whom were taking therapy for their previous conditions, while drugs administered during COVID-19 were reported in 55 out of 84 cases. The most common drugs included antibiotics (52.7%), antivirals (34.5%), hydroxychloroquine (32.7%), vasopressors (10, 18.1%), corticosteroids (12.7%), anticoagulants and heparins (12.7%) and biologics (10.9%). Diuretics, pain relievers and other medications were used in less than 5 patients (9%). The use of non-invasive ventilatory support was specified in 21 patients (38.1%), while 33 cases (60%) required intubation during hospitalization for COVID-19. Finally, 6 patients underwent hemodialysis (10.9%), while extracorporeal membrane oxygenation (ECMO) was used in 2 of them (3.6%).

The majority (73) of the deaths occurred in a hospital setting, i.e., intensive care or other wards, while among the 11 out of the hospital cases, 3 occurred in nursing homes, 5 patients were found dead in their homes, 1 in his car, while in 2 cases the data were not extractable. Post mortem examinations performed included complete autopsies (29 cases), partial autopsies (33 cases) and post mortem histology (22 cases). The histological samples involved the lung in all the 22 cases, heart in 10 cases, liver in 13, airways in 7, kidney in 2 and gastrointestinal tract in 2.

Imaging studies have been reported in 65 cases. Of these, in vivo imaging was performed in 50, post mortem in 14, both in 1. On in vivo radiographs, the most commonly reported features were bilateral patchy opacities and/or consolidations (60.7%), and 39.2% of chest CT showed ground glass opacity and/or consolidations. Post-mortem chest X-ray was performed in 2 cases and displayed bilateral opacities, while by PMCT, various degrees of pulmonary consolidation (80%), presence of reticular pattern (60%), pleural and/or pericardial effusions (40%) were described, as well as less represented features, e.g., emphysema, ground glass opacities and evidence of neoplastic lesions.

As for the autopsy room, macroscopic changes were described in several organs, although with a variable frequency, also caused by pre-existing pathologies. The most affected organs were lungs/airways in 51 of 54 cases, heart and vascular system in 33, liver in 12, kidneys in 11, spleen in 9, lymph nodes in 6 and CNS in 5 cases. Lungs were commonly described as heavy and edematous, with or without intraparenchymal hemorrhages or emboli. A macroscopic feature of pneumonia was also quite frequent, while purulent infections, empyema or green exudate were rarer. Extra-pulmonary common features included heart hypertrophy, though this is unlikely connected with COVID-19, enlargement of the spleen and of the lymph nodes. Alterations found in the gastrointestinal tract, prostate, skin, testis and other anatomical parts were much rarer. A similar picture was found in microscopic examinations of tissues, with lung/airways affected in all the 84 cases, liver in 40, heart and vascular system in 37, kidney in 25, spleen in 13, lymph nodes in 7, gastrointestinal tract in 3. Alterations reported in the CNS, bone marrow, testis and thyroid had lower frequencies. The most described finding within lung tissues was represented by diffuse alveolar damage (DAD) in exudative or organizing phases, coupled to pulmonary edema, hemorrhages and microthrombi. Less commonly, slight fibrosis, atypical pneumocytes or acute inflammatory infiltrates were noted. Microthrombi, together with signs of acute or chronic inflammation, were also reported in the trachea. Haemophagocytosis was occasionally noted in lymph nodes. In the heart, fibrosis and myocardiocyte hypertrophy have been mostly observed. In the liver, the dominant microscopic picture found was mild to severe hepatic steatosis, though portal/periportal inflammation, hepatocyte necrosis and hepatic congestion have been also described. The spleen commonly showed hyperplasia of the white pulp. In the kidney, arteriolosclerosis was the most frequently encountered finding, often related to chronic hypertensive damage and diabetes.

Regarding other examinations carried out, 9 electron microscopy tests were performed. In 8 of these 9 cases, the authors found viral-like particles within cells of different tissues (such as tracheal epithelial cells, pneumocytes, enterocytes, renal tubular cells). In the remaining case, no viral-like particles were found, but neutrophils in the alveolar capillaries and fibrin deposits in the alveolar spaces were documented. Moreover, the presence of bacteria, fungi or viruses in addition to SARS CoV-2 was documented in 7 of the patients by using cultural tests, rt-PCR and other laboratory tests. In 2 cases, toxicological investigations were also carried out, finding dextromethorphan in one patient (part of the antitussive therapy taken during COVID-19) and in another patient caffeine and naloxone. Causes of death were reported in 71 decedents, while the role of COVID-19 was specified 51 of them, being considered “cause of death” in 37 cases (72.5%), “contributing factor” in 12 (23.5%) and “significant factor” in 2 (3.9%).

A summary of the results is shown in Figure 2.

### 4.2. CSS Application

The COVID-19 Significance Score was applied to each case found in the literature review. The non-parametric ANOVA comparing the CSS assigned by three independent blinded investigators did not show significant differences (*p* > 0.05). Complete agreement was found in 68 cases. As shown in Table 2, 57 of the 84 reported deaths fell into CSS category 3 for at least 2 raters, which means that “COVID-19 was the leading cause of death”.

Twenty-two of the deaths fell into category 2 for at least two raters. In these cases, “COVID-19 likely contributed to the patient’s death, together with other factors that may have played a prominent role”. Four deaths were included in category 1, where “an alternative cause of death was likely”, by at least two raters. In one case, CSS was classified as U by two raters, further specific investigation being necessary, and as 0 for the third one.

### 4.3. Hamburg Score

The two classifications of deaths were compared in the cases reported in this work also described in the study by Edler et al., specifically in 13 of the 84 deaths collected in this paper (Table 2). Considering the Hamburg category 1 (defined COVID-19 death) equivalent to the CSS category 3, and the Hamburg category 2 (probable COVID-19 death) equivalent to the CSS category 2, the results of the two classification systems agreed in 8 out of the 13 cases. In the remaining 5 cases, differences in assessment emerged. Particularly:in cases 1, 3, 4, 12 by Wichmann et al. [40], CSS classified COVID-19 as the cause of death (CSS = 3), while the Hamburg score revealed a probable COVID-19 death (corresponding to CSS = 2);in case 2 by Wichmann et al. [40], CSS classified COVID-19 as the cause of death (CSS = 3), while Hamburg scored the fatality as possible COVID-19 death (corresponding to CSS = 1).

The *t*-test between the average CSS score and the Hamburg score, converted into CSS, did not yield a statistical significant difference.

## 5. Discussion

Information about 84 deaths involving SARS CoV-2 positiveness or infection have been collected in the present study, showing the growing interest of the literature with respect to post-mortem findings in COVID-19 pandemic.

So far, several tests have been developed in order to confirm a patient’s positivity to the virus, none of them free from issues of sensitivity [31]. The most used is certainly rt-PCR performed on swabs collected from the upper airways. All patients included in this study tested positive for airway swabs, performed in vivo or post-mortem. However, the accuracy of post-mortem swabs is yet to be clearly defined and false negative are theoretically possible [69]. Indeed, even though different studies report positive swabs even after several days, the influence of post-mortem interval and bacterial superimposition is unknown. One study suggested it might be reliable until 5 days [70]. C. Edler et al. verified the post-mortem sensitivity of the nasopharyngeal and oropharyngeal swabs by performing the test on 30 deceased at the time of dissection, finding a positive swab in 100% of cases, with a maximum time elapsed from death to the test of 12 days [38]. Furthermore, a study by Marco Dell’Aquila et al. highlighted the importance of performing multiple swabs in the post-mortem examination [71]. COVID-19 has been detected by nasopharyngeal and oropharyngeal swabs up to 27 h after death [72], while in another study the positivity of throat swabs lasted up to 128 h [73]. By performing rhino-pharyngeal, tracheal and lung swabs in 12 autopsy cases of patients with a clinical diagnosis of Sars-CoV-2 related pneumonia, 9 out of 12 cases had at least one post-mortem swab positive for Sars-CoV-2, with the virus found in samples up to 310 h from the post mortem sampling [71]. Moreover, a paper by Prema Seetulsingh et al. described the case of a patient who died of respiratory failure during transport to the hospital, with a negative upper airway swab, but with SARS CoV-2 found in the lung at an analysis performed 27 days following the death [74]. However, a correlation between the negativity of the lung swabs and the number of days elapsed from the ante mortem swabs was also found, as well as a negative correlation between the positivity of the other swabs and the number of days passed from the ante mortem swabs [71]. Thus, results of swabs should be interpreted with caution and never taken as an evidence of COVID-19 when singularly considered.

As a matter of fact, despite multiple reports allowed to assess the vitality of SARS CoV-2, scientific evidence regarding the risk of becoming infected for health care personnel arising from human dead hosts is lacking. Notwithstanding this, the risk of contagion involved in the post-mortem examination led some countries to discourage the performance of autopsies, as happened in Italy [75]. This might explain why the number of cases here considered, although significant, is rather low when compared to the high worldwide mortality for COVID-19. An additional possible explanation for the decline of the autopsy rate might be connected to the guidance for the safe management of a dead body, published by the World Health Organization and by the Center for Disease Control and Prevention [76,77]. Indeed, not all the autopsy facilities could be equipped with the required safety measures (e.g., negative pressure rooms) and the lack of “safe” autopsy rooms might have additionally led to a reduction in postmortem examinations [69,76].

The epidemiology of the victims, and the rate of comorbidities (absent in 28% of the cases), do not allow to confirm that SARS-CoV-2 is only affecting the elderly or patients who bear in already critical conditions. Rather, this is a confirmation that COVID-19 can be lethal even in healthy people and this should be taken in mind by forensic pathologists, who might incur in an otherwise unexplained death. As for the history of the disease, reported symptoms, laboratory alterations and macro as well as microscopic findings of the cases collected in this study were in line with those reported by other works, showing a prevalence of lung damage with edema, acute and late phase of DAD, presence of microthrombi in the pulmonary vessels or pneumonia [78,79,80,81,82,83], but also involvement of other organs, such as kidneys, heart and liver [84,85,86,87]. This also highlights that investigations limited to the lungs might not be enough to obtain a clear clinical post-mortem picture. Moreover, the complexity of the histological features shown even within the lungs might suggest that a biopsy-based approach might not be representative of the whole parenchyma.

As for the type of analyses performed, in vivo imaging was far more common than in the post-mortem setting (only 15 cases). Particularly, post-mortem imaging was performed when in vivo instrumental analysis was missing, e.g., cases 1,3,5,7–12 by Dominic Wichmann et al. [40], and its concordance or discordance with pathological findings allowed a high inter-rater agreement in the assignment of the CSS. Its application is strongly encouraged, especially when other info might be missing. Toxicology was extremely rarely applied. However, several drugs were administered before and during COVID-19 in most cases and, in this condition, it would appear reasonable to confirm the effectiveness of the administration, e.g., by excluding under- as well as over-doses.

A rather worrying picture emerged from the type of post-mortem examination performed, since the majority of cases (55 in total, out of 84) were not complete, nor performed with respect to the international guidelines [88,89]. Indeed, even though micro-invasive autopsies, especially if coupled to post-mortem imaging and extensive sampling of tissues for histology and electronic microscope-based analyses, might represent a viable alternative to reduce the risk of infection for health care personnel, the exclusion of some organs (most often, the brain) or the loss of a global view on the health status of the victim might lead to false conclusions. Especially in the case of such a widespread and systemic infection as COVID-19, which might affect multiple organs and lead to an unpredictable and severe immune response, the careful dissection of each organ appears of paramount importance. Indeed, a full autopsy is the only chance to observe the systemic changes and take optimal samples to identify the cause of death [28].

As already emerged for some toxicological issues, when the scientific data are scarce [35,36], a multidisciplinary evaluation is necessary and shared criteria might aid forensic pathologists in their delicate task, which has many consequences.

By observing the CSS applied to the collected cases, it can be noted that most of the deceased fall into the category “deaths from COVID-19”. A similar result is reported by an interesting study by Francesco Grippo et al. [5] By analyzing more than 5000 death certificates compiled according to the ICD, it was observed that COVID-19 was reported as the leading cause of death in 88% of cases. Sefer Elezkurtaj also confirms, by performing autopsies on 26 patients, that in the majority of decedents, the causes of death were directly related to SARS CoV-2 [90]. According to this study, the majority of patients had “died of COVID-19”, with only a contributory implication of pre-existing health conditions to the mechanism of death. However, the influence of a publication bias should be considered.

The very good agreement found by three blind and independent raters allows to hypothesize that the CSS is an easy tool which could be applied in the everyday routine of post-mortem examination on SARS-CoV-2 positive deceased, even by less experienced pathologists. Regarding the comparison with the Hamburg score by Edler et al. [38], as previously mentioned, it was not possible to apply that categorization in all cases, but only in the autopsies performed in Hamburg which were also reported in the studies by Wichmann et al. [40] and Heinrich et al. [57]. The study of the University of Hamburg, in fact, collected the key points of the first 80 consecutive autopsies carried out in the federal state of Hamburg, then applied a categorization of deaths on the basis of the causes of death reported, making this scale not usable in different studies. Furthermore, the 80 cases described by Edler et al. do not contain extractable information, particularly regarding post-mortem findings [38]. Therefore, they could not be included in our database of literature cases. The Hamburg score mostly considered the findings of “pneumonia”, “ARDS” and “pulmonary embolism” as indicators for a COVID-19-related death. Even if these findings are certainly fundamental in the evaluation of the role of SARS-CoV-2, we believe that a more comprehensive overview, as well as a valorization of past history and of the status of the other organs and functions (e.g., of coagulation), are needed. For example, findings of aspiration pneumonia in a patient with neurological comorbidities might be misinterpreted as SARS-CoV-2-related, when they might have occurred even in the absence of this pathogen and in any moment of the patient’s life (e.g., case 6 by Wichmann et al. [40], assigned CSS = 1). This further underlies the importance of collecting in vivo data when performing a post-mortem assessment. Clinical risk prediction models (e.g., QCOVID) have already been developed and validated on large population sets, to estimate the risk of becoming infected and then of dying of COVID-19, or of dying when admitted to hospital with COVID-19 [18,91,92].

The quantification of such risks might be certainly useful even in the postmortem setting, and might give forensic pathologists strong indications on the most important clinical predictors of death. However, these statistical ex ante tools do not allow one to assess ex post the cause of death and the role of the virus in the specific evaluated case. Thus, in the post-mortem evaluation, clinical stratification risk models or image-based outcome models should be always integrated with the CSS [93].

Even though the results, by comparing the converted Hamburg score and the CSS, were not significantly different, this type of analysis has been made possible only in a minority of cases and further studies would be needed to establish whether they are interchangeable. Nevertheless, we agree that, when the cause of death is difficult to be ascertained, a high degree of suspicion for COVID-19 should be maintained, and this probably had a reflection in the above-mentioned high degree of CSS 3 and 2 assigned.

Beside the difference in numbers among cases classified by Hamburg score and by CSS, the study has several limitations. Until July 31, only a few reports of complete autopsies had been published. The early publication of the present study has the aim to provide a quick overview and practical instruments which might be helpful for further cases evaluation. Despite the diffusion of safety protocols, very often these were not applied due to the infectious risk, preferring minimally invasive approaches such as ultrasound guided biopsies or partial autopsies, by opening of the thoracic and abdominal cavity, but leaving the organs in situ. Additionally, not all the articles reported information such as laboratory tests performed, comorbidities, circumstances of death and radiology. Regarding the swabs, all cases found were positive for the virus, but it has not always been reported whether the swab was performed in vivo or post mortem and, when performed post mortem, when with respect to the post-mortem interval. This information could be important in understanding how long the virus remains detectable in the patient’s airways after death, with implications in CSS, built to evaluate SARS CoV-2 positive patients. The lack of one or some of the CSS key points could make the score less accurate. A possible solution could consist in the creation of a register that contains all the autopsies performed on patients affected by COVID-19, with findings organized in a systematic way.

## 6. Limitations

The present systematic review has several limits. Firstly, the time of publication chosen was quite narrow, from the early months of 2020 to 31 July 2020. However, this was necessary due to the urgency of the matter. A broader period of observation would certainly provide more relevant data. Secondly, only papers at least providing some results of a post-mortem examination were included. This was motivated by the will of obtaining stronger evidence, even though we are aware that this might have resulted in a lower number of cases. Indeed, the total number of cases herein reported is certainly low, when compared to the worldwide mortality from Covid-19. This might be due to the limitations in performing autopsies which have been established, due to the infective risk for health care personnel and forensic pathologists, in many countries. Thirdly, the comparison between the CSS and the Hamburg score was only limited to a few cases. Finally, all relevant studies were included, with no distinction on the basis of the adherence to ethical standards and of the conflicts of interest, neither selecting only high-impact journal. This was done in order to offer a broad collection of cases, though it has resulted in the inclusion of a withdrawn article. However, the corresponding paper only provided a single case; thus, statistics were only minimally affected.

## 7. Conclusions

As the pandemic continues to claim victims, it is fundamental to distinguish those patients who have died “from COVID-19” from those who have died “with COVID-19”. The SARS CoV-2 Significance Score (CSS) used after a complete accurate post-mortem examination, coupled to the retrieval of in vivo data, post-mortem radiology, histology and toxicology, as well as to additional required analyses (e.g., electronic microscopy) aims to be a useful, concise tool helping in the assessment of the cause of death and the role played by this virus. A shared use of this scale might hopefully lower the inhomogeneities in forensic evaluation of SARS-CoV-2.

## Figures and Tables

**Figure 1 diagnostics-11-00190-f001:**
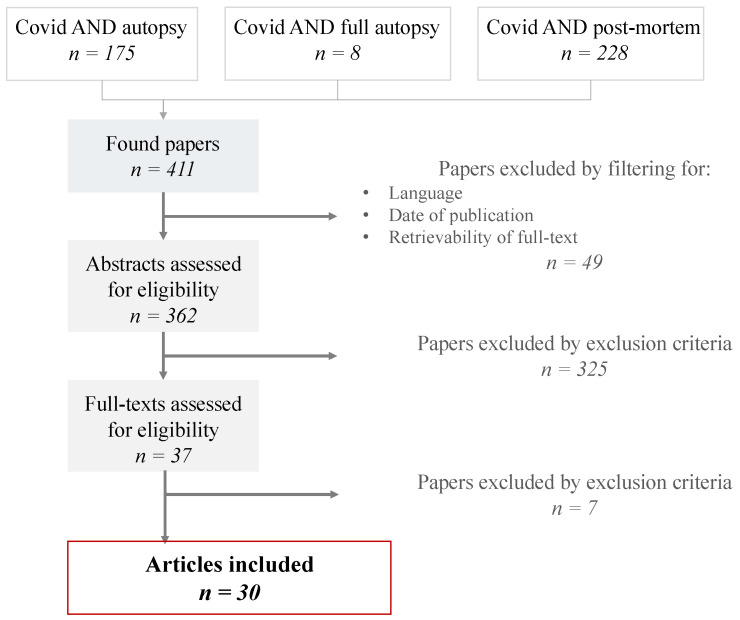
PRISMA flow chart of the selection process.

**Figure 2 diagnostics-11-00190-f002:**
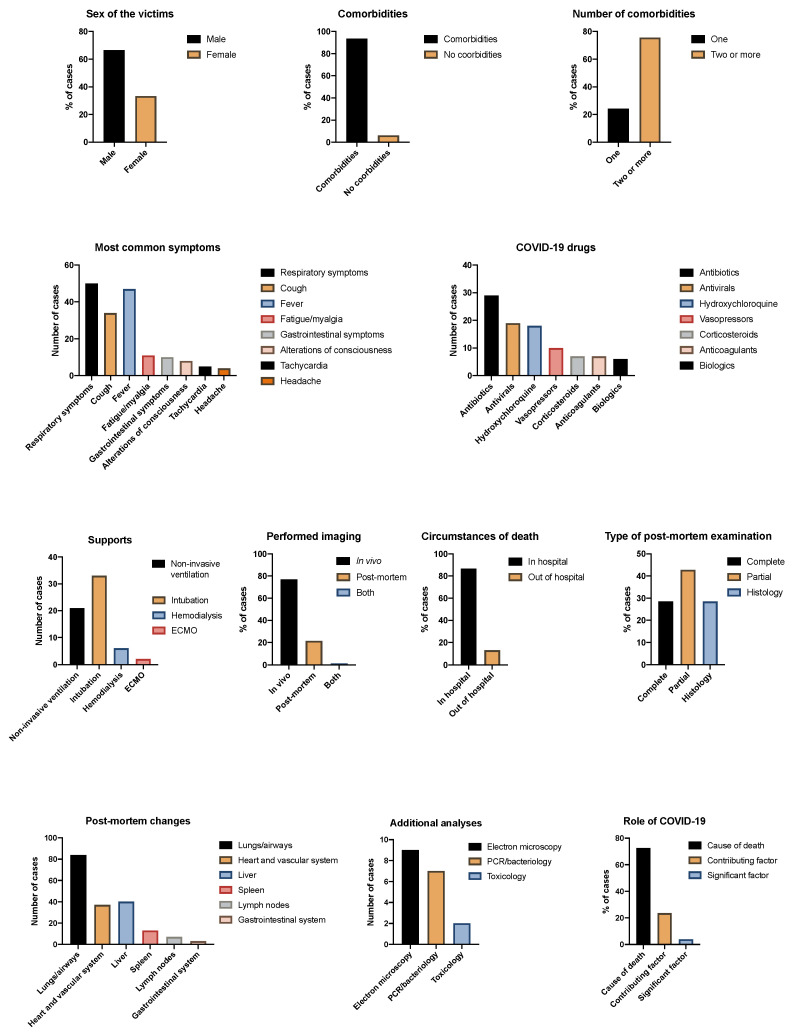
Summary of the main in vivo and post-mortem data emerged from literature cases.

**Table 1 diagnostics-11-00190-t001:** Detailed results of the literature cases.

Author	A	G	IN VIVO DATA					POSTMORTEM DATA						
			Comorbidities and Past Drugs	Therapy	Labor	Imaging	Course of the Disease/Circumstances of Death	Imaging	Macroscopic Features	Microscopic Features	Tox	Additional Analyses	Cause of Death	Swabs
Benjamin T Bradley et al.	57	M	CKD, DM2, HTN, OSAS, obesity	Intubation	elevated creatinine, lymphocytopenia	Chest x-ray: bilateral multifocal patchy airspace opacities	Hospital presentation: 4-day history of cough, fever, chills, fecal incontinence, fatigue and onset of respiratory distress. Intubated, died 6 days after admission	N/D	LUNGS: heavy and edematous	LUNGS: Pulmonary edema, acute phase DAD, multinucleated giant cells, reactive pneumocytes HEART: Interstitial fibrosis, myocyte hypertrophy LIVER: Steatosis, periportal lymphocytic inflammation KIDNEY: Moderate to severe arterionephrosclerosis, diabetic changes, scattered tubular casts TRACHEA: Edema, chronic (lymphocytic) tracheitis GI: Multifocal gastric hemorrhages	N/D	N/D	(ICD-10 code) A: Coronavirus Disease 2019 (COVID-19) pneumonia OSC: DM, end stage CKD, HTN	Positive for SARS CoV-2 (unspecified)
	74	F	DM2, OSAS, AF, pulmonary hypertension, CKD, obesity	Intubation	elevated creatinine	Chest x-ray: increase in vascular and interstitial opacities	Hospital presentation: 2-day history of AKI, delirium, cough, acute cardiomyopathy and respiratory distress. Intubated, died on the day of admission	N/D	N/D	LUNGS: Organizing phase DAD, reactive pneumocytes, acute bronchiolitis, alveolar septal thickening HEART: Interstitial fibrosis, myocyte hypertrophy, replacement fibrosis LIVER: Steatosis, congestion KIDNEY: Moderate to severe arterionephrosclerosis, diabetic changes TRACHEA: Edema, chronic (lymphocytic) tracheitis	N/D	N/D	(ICD-10 code) A: cardiomyopathy B: COVID-19 OSC: DM, pulmonary hypertension, immunosuppression	Positive for SARS CoV-2 (unspecified)
	54	M	Neurological alteration and dysphagia from previous head injury	N/D	N/D	Chest x-ray: bilateral patchy opacities	Hospital presentation: 1-day history of fever, respiratory distress and tachycardia. Refused intubation and died the day after admission	N/D	N/D	LUNGS: Pulmonary edema, reactive pneumocytes, acute bronchiolitis, background emphysematous change, microthrombi HEART: Interstitial fibrosis, myocyte hypertrophy LIVER: Periportal lymphocytic inflammation, centrilobular necrosis KIDNEY: Mild arterionephrosclerosis, scattered tubular casts TRACHEA: Acute neutrophilic tracheitis, fibrosis and ossification, microthrombi	N/D	N/D	(ICD-10 code) A: aspiration pneumonia and sepsis B: COVID-19 OSC: dysphagia due to traumatic neurological damage	Positive for SARS CoV-2 (unspecified)
	74	M	Heart failure with preserved EF, frontotemporal dementia, HTN, OSAS	Intubation	N/D	Chest x-ray: diffuse bilateral scattered opacities	Hospital presentation: cough, myalgia, respiratory distress and fever. Intubated, died on the day of admission	N/D	N/D	LUNGS: Pulmonary edema, acute phase DAD, multinucleated giant cells, reactive pneumocytes, alveolar septal thickening, patchy perivascular lymphocytic inflammation HEART: Interstitial fibrosis, myocyte hypertrophy, replacement fibrosis LIVER: Steatosis, congestion, features of toxic or metabolic disease KIDNEY: Mild to moderate arteriolosclerosis, scattered tubular casts TRACHEA: Acute neutrophilic tracheitis	N/D	N/D	(ICD-10 code) A: ARDS B: viral pneumonia C: COVID-19 OSC: CKD	Positive for SARS CoV-2 (unspecified)
	73	F	DM2, HTN, congestive heart failure, hypothyroidism, obesity, schizoaffective disorder, bipolar disorder	Intubation	N/D	Chest x-ray: widespread bilateral opacities	Hospital presentation: 5-day history of cough, respiratory distress and fever. Intubated, died 8 days after admission	N/D	N/D	LUNGS: Pulmonary edema, acute phase DAD, multinucleated giant cells, alveolar septal thickening, perivascular and interstitial lymphocytic inflammation HEART: Interstitial fibrosis, myocyte hypertrophy LIVER: Steatosis, congestion, lobar neutrophilic inflammation KIDNEY: Mild arterionephrosclerosis, scattered tubular casts, diabetic changes SPLEEN: White pulp depletion TRACHEA: Edema, chronic (lymphocytic) tracheitis	N/D	N/D	(ICD-10 code) A: ARDS B: viral pneumonia C: COVID-19 OSC: obesity, HTN, DM	Positive for SARS CoV-2 (unspecified)
	84	F	COPD, congestive heart failure, AF, aortic stenosis, HTN, CKD, osteoporosis	N/D	lymphocytopenia	Chest x-ray: bibasilar atelectasis or consolidations with small pleural effusions	Hospital presentation: 1-day history of respiratory distress and delirium. Refused intubation and died the day after admission	N/D	LUNGS: presence of intraparenchymal hemorrhages	LUNGS: Acute phase DAD, reactive pneumocytes, pulmonary hemorrhage, acute bronchopneumonia, background emphysematous changes HEART: Interstitial fibrosis, myocyte hypertrophy LIVER: Congestion, portal lymphocytic inflammation KIDNEY: Mild to moderate arterionephrosclerosis, scattered tubular casts TRACHEA: Edema, chronic (lymphocytic) tracheitis	N/D	N/D	(ICD-10 code) A: ARDS B: viral pneumonia C: COVID-19 OSC: COPD, AF, aortic stenosis, mitral stenosis	Positive for SARS CoV-2 (unspecified)
	71	M	HTN, dyslipidemia, coronary heart disease, AF, CKD, OSAS	N/D	elevated creatinine, lymphocytopenia	Chest x-ray: bilateral multifocal opacities	Hospital presentation: 7-day history of cough and respiratory distress. *Pseudomonas aeruginosa* found in the sputum. Refused intubation and died 6 days after admission	N/D	N/D	LUNGS: Pulmonary edema, acute and organizing DAD, reactive pneumocytes, alveolar septal thickening, pulmonary hemorrhage HEART: Interstitial fibrosis, myocyte hypertrophy, vascular predominant amyloid LIVER: Congestion KIDNEY: Severe arterionephrosclerosis, vascular predominant amyloid TRACHEA: Edema, acute (neutrophilic) tracheitis	N/D	P. aeruginosa found in the sputum	(ICD-10 code) A: viral pneumonia B: COVID-19 C: immunosuppression OSC: end-stage CKD, coronary heart disease, stroke	Positive for SARS CoV-2 (unspecified)
	76	F	Dyslipidemia, osteoporosis	Intubation	elevated creatinine, lymphocytopenia, elevated troponin	Chest x-ray: bilateral multifocal opacities	Hospital presentation: 3-day history of respiratory distress, hypotension, tachycardia and fever. *Staphylococcus aureus* e Virus influenza A detected. Intubated, died 4 days after admission	N/D	LUNGS: heavy and edematousSPLEEN: splenomegalyCNS: scattered punctate subarachnoid hemorrhages	LUNGS: Pulmonary edema, acute and organizing phase DAD, reactive pneumocytes, multinucleated cells, alveolar septal thickening, acute bronchiolitis, perivascular and interstitial lymphocytic inflammation, microthrombi HEART: Interstitial fibrosis, myocyte hypertrophy, replacement fibrosis, myocarditis LIVER: Steatosis, centrilobular necrosis RENE: Mild arterionephrosclerosis, scattered tubular casts, reactive tubular epithelium, chronic (lymphocytic) interstitial inflammation SPLEEN: Splenic infarction TRACHEA: Edema, chronic (lymphocytic) tracheitis SUBCARINAL LYMPH NODE: Rare haemophagocytosis CNS: Punctate subarachnoid hemorrhages	N/D	Methicillin-resistant S. aureus and influenza A virus. SARS CoV-2 RNA detected in multiple organs. ELECTRON MICROSCOPY: viral-like particles in tracheal epithelial cells, pneumocytes, enterocytes, proximal convoluted tubule cells, large intestine	(ICD-10 code) A: ARDS B: viral pneumonia C: COVID-19 OSC: influenza A, staphylococcal pneumonia, myocarditis, cardiomyopathy, septic shock	Positive for SARS CoV-2 (unspecified)
	75	F	Dyslipidemia, DM2, coronary heart disease, congestive heart failure, CKD, COPD, DVT	N/D	lymphocytopenia	Chest x-ray: bilateral interstitial opacities, asymmetric edema on the right	The patient presented to the hospital with a 3-day history of delirium, fever and respiratory distress. She refused intubation and died 9 days after admission	N/D	N/D	LUNGS: Edema, acute phase DAD, reactive pneumocytes, acute bronchiolitis, microthrombi HEART: Interstitial fibrosis, myocyte hypertrophy LIVER: Steatosis, congestion KIDNEY: Moderate to severe arterionephrosclerosis SPLEEN: White pulp depletion TRACHEA: Edema, chronic (lymphocytic) tracheitis, microthrombi	N/D	N/D	(ICD-10 code) A: ARDS B: pneumonia C: COVID-19 OSC: CKD, DM, thromboembolism	Positive for SARS CoV-2 (unspecified)
	84	M	CKD, COPD, dyslipidemia, OSAS, mitral regurgitation, complete AV block, chronic pain, arthritis, obesity, HTN	N/D	elevated creatinine, lymphocytopenia	Chest x-ray: complete opacification of the left hemithorax, opacities in the right middle and lower lobes	Hospital presentation: 1-day history of delirium, hypotension and respiratory distress. Refused intubation and died the same day of admission	N/D	LUNGS: heavy and edematous, subsegmental emboli	LUNGS: Pulmonary edema, focal acute phase DAD, reactive pneumocytes, acute and chronic bronchitis, perivascular and interstitial lymphocytic inflammation, background emphysematous changes, subsegmental pulmonary embolus HEART: Interstitial fibrosis, myocyte hypertrophy, replacement fibrosis LIVER: Congestion KIDNEY: Mild to moderate arterionephrosclerosis, reactive tubular epithelium TRACHEA: Edema, chronic (lymphocytic) tracheitis	N/D	N/D	(ICD-10 code) A: hypoxemic and hypercapnic respiratory failure B: pulmonary emphysema OSC: COVID-19, HTN, mitral regurgitation, stage 3 CKD	Positive for SARS CoV-2 (unspecified)
	81	F	HTN, dyslipidemia, breast cancer, CKD, demyelinating neuropathy, lacunar infarcts, recent pneumonia, Alzheimer’s disease	Intubation	elevated troponin, lymphocytopenia	Chest x-ray: bilateral multifocal opacities	Hospital presentation: 1-day history of fever, cough nausea and vomit. Intubated after 4 days, died 6 days after admission	N/D	LUNGS: heavy and edematous	LUNGS: Acute and organizing diffuse DAD, reactive pneumocytes, multinucleated giant cells, acute bronchopneumonia, pulmonary hemorrhage HEART: Interstitial fibrosis LIVER: Steatosis, congestion KIDNEY: Mild to moderate, arterionephrosclerosis, scattered tubular casts TRACHEA: Edema, acute (neutrophilic) tracheitis SUBCARINAL LYMPH NODE: Haemophagocytosis	N/D	N/D	(ICD-10 code) A: hypoxemic respiratory failure B: ARDS C: viral and bacterial pneumoniaD: COVID-19 OSC: HTN	Positive for SARS CoV-2 (unspecified)
	42	F	History of lobular breast cancer with bilateral mastectomy and neoadjuvant chemotherapy	Intubation	leukocytosis, lymphocytopenia	Chest x-ray: bilateral multifocal opacities	Hospital presentation: 5-day history of fever and headache. Intubated after 7 days, died 9 days after admission	N/D	LUNGS: heavy and edematous, subsegmental emboli	LUNGS: Pulmonary edema, acute and organizing phase DAD, reactive pneumocytes, multinucleated giant cells, acute bronchiolitis, subsegmental pulmonary emboli HEART: Interstitial fibrosis, myocyte hypertrophy, replacement fibrosis LIVER: Steatosis, congestion, centrilobular necrosis KIDNEY: Mild to moderate arteriolosclerosis, scattered granular casts SPLEEN: White pulp depletion TRACHEA: Edema	N/D	N/D	(ICD-10 code) A: ARDS B: COVID-19 OSC: adjuvant therapy for breast cancer	Positive for SARS CoV-2 (unspecified)
	71	M	Coronary heart disease, ischemic cardiomyopathy, HTN, aortic stenosis, end-stage CKD, chronic pulmonary fibrosis, history of cerebellar stroke	N/D	elevated creatinine, elevated troponin, lymphocytopenia	Chest x-ray: reduced lung volumes, diffuse pulmonary changes, compatible with pulmonary fibrosis	Hospital presentation: 1 day-history of shortness of breath, bradycardia, new onset AV block and delirium. Worsening hypoxia, refused intubation. 4 days after hospitalization, died of cardiac arrest	N/D	LUNGS: heavy and edematousSPLEEN: splenomegaly	LUNGS: Pulmonary edema, acute phase DAD, pulmonary hemorrhage, chronic fibrosis, microthrombi HEART: Interstitial fibrosis, myocyte hypertrophy, replacement fibrosis, myocardial amyloid LIVER: Congestion KIDNEY: Severe arterionephrosclerosis, scattered tubular casts, reactive tubular epithelium, renal vein organizing thrombuse TRACHEA: Sloughed epithelium	N/D	SARS CoV-2 RNA detected in multiple organs. ELECTRON MICROSCOPY: viral-like particles in tracheal epithelial cells, pneumocytes, enterocytes, proximal convoluted tubule cells, large intestine	(ICD-10 code) A: ventricular fibrillation B: ARDS C: COVID-19 OSC: cardiac conduction system anomalies, cardiac amyloidosis, ischemic heart disease, HTN, pulmonary fibrosis, end-stage CKD, cervical spinal stenosis	Positive for SARS CoV-2 (unspecified)
	73	F	HTN, asthma, DM, dyslipidemia, obesity	intubation	elevated LDH and leukocytosis	Chest x-ray: reduced lung volume, diffuse bilateral changes	Hospital presentation: 2 day-history of progressive shortness of breath and respiratory distress. Physical examination revealed hypoxia and signs of shock. Developed multifactorial encephalopathy, AF and presumed ventilator-associated pneumonia. Intubated, died 21 days after hospitalization	N/D	LUNGS: heavy and edematous. Parenchymal consolidations	LUNGS: Pulmonary edema, acute bronchopneumonia, perivascular and interstitial lymphocytic infiltrate, microthrombi, reparative fibrosis and neovascularisation, vascular disease HEART: Interstitial fibrosis, myocyte hypertrophy, replacement fibrosis LIVER: Steatosis, congestion, centrilobular necrosis, portal lymphocytic inflammation KIDNEY: Mild to moderate arterionephrosclerosis, reactive tubular epithelium, tubular casts, chronic inflammation, focal segmental glomerulosclerosis TRACHEA: Edema, chronic (lymphocytic) tracheitis, hemorrhage, ulceration, epithelial sloughing SUBCARINAL LYMPH NODE: Haemophagocytosis	N/D	SARS CoV-2 RNA detected in multiple organs	(ICD-10 code) A: ARDS B: COVID-19 OSC: HTN, asthma, DM2, AF, obesity	Positive for SARS CoV-2 (unspecified)
Dominic Wichmann et al.	52	M	Obesity (BMI 38.8 kg/m^2^)	N/D	N/D	N/D	N/D	CT: diffuse bilateral pulmonary consolidations	LUNGS: pulmonary embolism, pneumonia HEART AND VESSELS: cardiomegaly, DVT, atherosclerosis OTHERS: splenomegaly, hepatomegaly, shock organs (liver, kidneys)	LUNGS: DAD, reactive pneumocytes, fibroblasts, giant cells, scattered hyaline membranes, slight fibrosis, congestion of small vessels,	N/D	N/D	CLINICAL: sudden cardiac death PATHOLOGICAL: pulmonary embolism, pneumonia	POST MORTEM: positive for SARS CoV-2 (nasopharyngeal)
	70	M	Parkinson’s disease, coronary heart disease, PVD, CKD	Rivaroxaban, piperacillin/tazobactan	Elevated LDH, elevated creatinine, elevated CRP	N/D	N/D	N/D	LUNGS AND AIRWAYS: pneumonia, purulent bronchitis HEART: coronary heart disease, signs of previous AMI, cardiomegalyOTHERS: muscle stiffness, shock liver	LUNGS: DAD, activated pneumocytes, hyaline membranes, scattered lymphocytes. Focal granulocyte infiltrates, acute and chronic bronchitis	N/D	N/D	CLINICAL: respiratory failure, pneumonia PATHOLOGICAL: pneumonia with bronchopneumonia	POST MORTEM: positive for SARS CoV-2 (nasopharyngeal)
	71	M	HTN, smoking, granulomatous pneumonia, obesity (BMI 36.8 kg/m2)	Vasopressors, intubation, meropenem, levofloxacin, enoxaparin	Elevated aPTT, Elevated LDH, Elevated CRP, Elevated creatinine	N/D	N/D	CT: emphysema, subtle reticular pattern in each lobe, consolidations in the lower right and lower left lobe	LUNGS: pulmonary embolism, pneumonia HEART AND VASES: coronary heart disease, DVT, atherosclerosis OTHERS: anasarca	LUNGS: DAD, squamous metaplasia, fibroblasts, hyaline membranes, activated pneumocytes, thromboemboli HEART: lymphocytic myocarditis in the right ventricle	N/D	N/D	CLINICAL: respiratory failure, pneumonia PATHOLOGICAL: pulmonary embolism, pneumonia	POST MORTEM: positive for SARS CoV-2 (nasopharyngeal)
	63	M	DM2, obesity (BMI 37.3 kg/m^2^), asthma	Vasopressors, intubation, cefpodoxime	Elevated D-dimer, Elevated LDH, Elevated CRP	N/D	N/D	N/D	LUNGS: pulmonary embolism, pneumonia HEART AND VESSELS: cardiomegaly, DVT OTHERS: ischemic colitis, liver in shock	LUNGS: DAD, fibroblasts, activated pneumocytes, hyaline membranes, squamous metaplasia, hemorrhagic infarcts, thromboemboli	N/D	N/D	CLINICAL: cardiopulmonary failure, pulmonary embolism PATHOLOGICAL: pulmonary embolism, pneumonia	POST MORTEM: positive for SARS CoV-2 (nasopharyngeal)
	66	M	Coronary heart disease	N/D	N/D	N/D	N/D	CT: consolidations in each lobe, reticular pattern in the upper and lower right lobes and in both left lobes	LUNGS: evidence of pneumonia HEART AND VESSELS: coronary heart disease, previous AMI, DVT	LUNGS: DAD, activated pneumocytes, fibroblasts, hyaline membranes, necrosis, lymphocytes, thromboemboli	N/D	N/D	CLINICAL: sudden cardiac death PATHOLOGICAL: pneumonia	POST MORTEM: positive for SARS CoV-2 (nasopharyngeal)
	54	F	Dementia, epilepsy, trisomy 21	N/D	Elevated LDH, Elevated CRP	N/D	N/D	CT: multiple right and left consolidations, ground glass opacities in the right lobes and in the upper left lobe, reticular pattern	LUNGS: pneumonia OTHERS: renal infarction, PEG	LUNGS: extensive granulocytic infiltrate in alveoli and bronchi, resembling focal bacterial bronchopneumonia. Acute bronchitis, congestion of small vessels	N/D	N/D	CLINICAL: respiratory failure, aspiration pneumonia PATHOLOGICAL: pneumonia	POST MORTEM: positive for SARS CoV-2 (nasopharyngeal)
	75	F	AF, smoking, coronary heart disease	O_2_, edoxaban	Elevated aPTT, Elevated D-dimer, Elevated LDH, Elevated CRP	N/D	N/D	CT: reticular pattern in each lobe, small areas of consolidation in the lower right lobe and both left lobes	LUNGS: pneumonia, pulmonary emphysema HEART AND VESSELS: coronary heart disease, left cardiac dilatation, mitral calcifications, cardiac pacemaker, atherosclerosis	LUNGS: DAD, hyaline membranes, activated pneumocytes, squamous metaplasia, emphysema, small vessel congestion	N/D	N/D	CLINICAL: respiratory failure, viral pneumonia PATHOLOGICAL: pneumonia	POST MORTEM: positive for SARS CoV-2 (nasopharyngeal)
	82	M	Parkinson’s disease, DM2, coronary heart disease	N/D	Elevated D-dimer, Elevated LDH, Elevated CRP	N/D	N/D	CT: emphysema, diffuse consolidation in each lobe, reticular pattern in the upper and lower right lobes and in the lower left lobe. Bilateral pleural effusions	LUNGS: pneumonia, emphysema HEART AND VESSELS: coronary heart disease, previous AMI with left heart aneurysm, atherosclerosis, DVT	LUNGS: extensive granulocytic infiltrate in alveoli and bronchi, resembling focal bacterial bronchopneumonia, emphysema	N/D	N/D	CLINICAL: respiratory failure, viral pneumonia PATHOLOGICAL: bronchopneumonia	POST MORTEM: positive for SARS CoV-2 (nasopharyngeal)
	87	F	Pulmonary NET, COPD, coronary heart disease, CKD	N/D	Elevated CRP	N/D	N/D	CT: emphysema, spherical tumor in the lower right lobe, small areas of consolidation in the upper and lower right lobes and in the upper left lobe, reticular pattern in the upper and lower right lobes and both left lobes	LUNGS: pneumonia, purulent bronchitis, bullous emphysema, pulmonary NET HEART: coronary heart disease, previous AMI OTHERS: cachexia, atherosclerosis	LUNGS: extensive granulocytic infiltrate in alveoli and bronchi, resembling focal bacterial bronchopneumonia. Presence of emphysema, acute bronchitis, small cell NET	N/D	N/D	CLINICAL: respiratory failure, viral pneumonia PATHOLOGICAL: suppurative bronchitis	POST MORTEM: positive for SARS CoV-2 (nasopharyngeal)
	84	M	DM2, HTN, UC	N/D	Leukocytosis, elevated D-dimer, elevated LDH, elevated creatinine, elevated CRP	N/D	N/D	CT: reticular pattern in the upper and lower right lobes and in both the left lobes, consolidation in the middle and lower right lobes and in both the left lobes, ground glass opacities in the upper and middle right lobes and in a portion of the upper left lobe. Bilateral pleural effusions	LUNGS: pneumonia, emphysema HEART: previous IMA OTHERS: septicemia, atrophic kidneys	LUNGS: extensive granulocytic infiltrate in alveoli and bronchi, resembling focal bacterial bronchopneumonia. Emphysema, congestion of small vessels, chronic bronchitis, fibrosis	N/D	N/D	CLINICAL: respiratory failure, viral pneumonia PATHOLOGICAL: pneumonia, septic encephalopathy	POST MORTEM: positive for SARS CoV-2 (nasopharyngeal)
	85	M	Coronary heart disease, HTN, asthma, AF	Vasopressors, intubation, dialysis	Elevated aPTT, elevated D-dimer, elevated LDH, elevated CRP, elevated procalcitonin	N/D	N/D	CT: diffuse consolidations in each lobe, reticular pattern in the middle and lower right lobes and both left lobes, ground glass opacities in the upper and middle right lobes and in the upper left lobe. Bilateral pleural effusions	LUNGS: pneumonia, minor pulmonary embolism, emphysema HEART AND VESSELS: coronary heart disease, cardiomegaly, atherosclerosis, DVT	LUNGS: DAD, scattered hyaline membranes, giant cells, activated pneumocytes, emphysema, small vessel congestion, granulocyte infiltrates	N/D	N/D	CLINICAL: cardiac arrest due to respiratory failure PATHOLOGICAL: pneumonia	POST MORTEM: positive for SARS CoV-2 (nasopharyngeal)
	76	M	Obesity (BMI 34.4 kg/m^2^)	Vasopressors, intubation, certainparin	Elevated LDH, elevated CRP	N/D	N/D	CT: no ventilated area in both lungs, except for a small area in the upper and middle right lobes and in both the left lobes. Bilateral pleural effusions	LUNGS: pulmonary embolism with pulmonary infarcts, pneumonia, purulent tracheobronchitis, emphysemaHEART AND VESSELS: cardiomegaly, DVT	LUNGS: DAD, hyaline membranes, fibrosis, activated pneumocytes, lymphocytes, thrombosis, small vessel congestion, plasma cells, hemorrhagic infarcts	N/D	N/D	CLINICAL: pulmonary embolism PATHOLOGICAL: pulmonary embolism, respiratory infection	POST MORTEM: positive for SARS CoV-2 (nasopharyngeal)
Andrey Prilutskiy et al.	72	M	N/D	Azithromycin, HCQ, anakinra, intubation	hypertriglyceridemia, elevated ferritin, elevated CRP	N/D	Hospital presentation: a 4-day history of fever and progressive hypoxia. Intubated on 7th day, died 18 days after hospitalization	N/D	Enlarged mediastinal and lung lymph nodes	LUNGS: DAD in exudative phase. Mediastinal and pulmonary lymph nodes containing clusters of haemophagocytes, marked distension of the cortical and subcortical sinuses and focal necrosis. Lymphocytic depletion in the lymph nodes SPLEEN: White pulp depletion, red pulp infarction, histiocyte hyperplasia and hemosiderin-laden macrophages, suggestive of previous haemophagocytosis LIVER: mild centrilobular congestion, mild steatosis	N/D	immunohistochemistry for HHV8, CMV and EBER for EBV in lymph nodes with haemophagocytosis: negative H-score for haemophagocytic lymphohistiocytosis: 217 (HLH present)	ARDS, HLH	Positive for SARS CoV-2 (nasopharyngeal)
	91	M	N/D	Azithromycin, doxycycline, HCQ	elevated fibrinogen, elevated ferritin, elevated CRP	N/D	Hospital presentation: 1-day history of fever and progressive hypoxia. Refused intubation and died 8 days after admission	N/D	Enlarged mediastinal and pulmonary lymph nodes SPLEEN: enlarged, with a soft and crumbly appearance	LUNGS: signs of exudative DAD. Cluster of haemophagocytes in the mediastinal and pulmonary lymph nodes SPLEEN: large bleeding areas in red pulp, focal hemophagocytosis, white pulp depletion LIVER: mild centrilobular congestion, mild steatosis	N/D	immunohistochemistry for HHV8, CMV and EBER for EBV in lymph nodes with haemophagocytosis: negative H-score for haemophagocytic lymphohistiocytosis: 145 (incomplete score, triglyceridemia values were missing, probable HLH)	ARDS	Positive for SARS CoV-2 (nasopharyngeal)
	72	M	N/D	Azithromycin, ceftriaxone, sarilumab	increase in platelets, elevated fibrinogen, elevated CRP	N/D	Hospital presentation: 3-day history of fever and progressive hypoxia. Refused intubation and died 6 days after admission	N/D	Enlarged mediastinal and lung lymph nodes	LUNGS: exudative DAD. Mediastinal and pulmonary lymph nodes containing clusters of haemophagocytes SPLEEN: slightly hyperplastic white pulp, congestion of the red pulp LIVER: mild centrilobular congestion, mild steatosis BONE MARROW: myeloid hyperplasia	N/D	immunohistochemistry for HHV8, CMV and EBER for EBV in lymph nodes with haemophagocytosis: negative H-score for haemophagocytic lymphohistiocytosis: 131 (HLH absent)	ARDS	Positive for SARS CoV-2 (nasopharyngeal)
	64	F	N/D	Sarilumab, ceftriaxone, intubation	hypertriglyceridaemia, elevated fibrinogen, elevated ferritin, elevated CRP	N/D	Hospital presentation: 5-day history of fever and progressive hypoxia. Intubated, died 15 days after hospitalization	N/D	N/D	LUNGS: DAD in the exudative phase SPLEEN: hyperplastic white pulp, congestion of the red pulp LIVER: mild centrilobular congestion, mild steatosis BONE MARROW: myeloid hyperplasia	N/D	H-score for haemophagocytic lymphohistiocytosis: 96 (HLH absent)	ARDS	Positive for SARS CoV-2 (nasopharyngeal)
Hans Bösmüller et al.	78	F	Obesity (BMI 35.2 Kg/m^2^), HTN, AV block treated with permanent dual chamber pacemaker	N/D	N/D	N/D	Death at home after 12 h of fever, cough and vomiting	N/D	LUNGS: significant pulmonary edema, slight increase in the consistency of the lower lobes. HEART: 520 g, dilation of both ventricles	LUNGS: generalized edema. Capillary endothelitis with increased neutrophils. Microthrombi in alveolar capillaries and small pulmonary vessels (including septal veins). Focal inflammatory exudate with scattered neutrophils and hyaline membranes, with initial organizational changes LIVER: moderate acute congestion and activation of Kuppfer cells	N/D	qRT-PCR for cytokines in lung tissue revealed a massive increase in IL-1β mRNA and IL-6 mRNA. SARS CoV-2 RNA detected in the lungs	Early stage pneumonia with thrombotic microangiopathy, pulmonary edema and acute heart failure	Positive for SARS CoV-2 (pharyngeal)
	78	M	Coronary heart disease, HTN, DM, Parkinson’s disease	Anticoagulants, vasopressors, intubation	Lymphocytopenia, elevated D-dimer, elevated fibrinogen, elevated CRP, elevated IL-6, elevated LDH, elevated creatine kinase and ferritin. Progressive thrombocytopenia, terminal reduction of D-dimers and IL-6	N/D	Hospital presentation: 3 weeks of generalized weakness, fever and dry cough, worsening in the 3 days prior to admission. Intubated. After a general improvement, massive increase of D-dimers and IL-6, thrombocytopenia, MOF and shock. Died 4 days after the peak of D-dimers	N/D	LUNGS: significant pulmonary edema and consolidations. Macroscopically visible thrombi, especially in small to medium sized pulmonary vessels (both venous and arterial), areas of recent infarction OTHERS: moderate hepato-splenomegaly	LUNGS: Diffuse DAD with massive intra-alveolar fibrin deposits and hyaline membranes. Marked hyperplasia and desquamation of the alveolar epithelium. Diffuse areas of organized DAD with proliferation of fibroblasts and collagen fiber deposition in intra alveolar exudate. Focal massive presence of leukocytes in medium-sized vessels LIVER: signs of haemophagocytosis	N/D	ELECTRON MICROSCOPY: viral-like particles in lung endothelial cells and type 1 pneumocytes Blood cultures for bacteria and fungi: negative. SARS CoV-2 RNA detected in the lungs	ARDS, vasogenic shock and liver failure	Positive for SARS CoV-2 (pharyngeal)
	72	M	Coronary heart disease, HTN, rheumatic polymyalgia, history of Merkel cell carcinoma (adjuvant radiotherapy in progress)	Meropenem, dialysis, intubation	Lymphocytopenia, elevated CRP, elevated IL-6. 6 days after admission leukocytosis, elevated D-dimer, elevated CRP, elevated IL-6, elevated procalcitonin	N/D	Hospital presentation: syncope, fever, cough and vomiting. Intubated 4 days after admission. Acute hypercapnia and several laboratory changes arose on day 6. Diagnosis of Klebsiella oxytoca pulmonary superinfection. Liver and kidney failure, with death due to liver failure 10 days after hospitalization, despite dialysis	N/D	LUNGS: macroscopic picture similar to patient 2, with macroscopically visible thrombi in the pulmonary vessels and consolidations in both lower lobes OTHERS: Hepato-splenomegaly. The surface of the liver was yellowish and dark	LUNGS: advanced DAD, with extensive hyaline membranes and concentration of intra-alveolar macrophages, multiple giant cells and pronounced hyperplasia of the alveolar epithelium (partly atypical). Focal squamous metaplasia and areas of organizing pneumonia. Viral particles in the endothelial cells of the lung capillaries and in the interstitial spaces	N/D	blood cultures for bacteria and fungi: negative. After 6 days of hospitalization, Klebsiella oxytoca superinfection. SARS CoV-2 RNA detected in the lungs	ARDS; vasogenic shock, liver failure	Positive for SARS CoV-2 (pharyngeal)
	59	M	Obesity (BMI 35.8 Kg/m^2^), asthma, HTN	ECMO, dialysis	Elevated D-dimer	N/D	Hospital presentation: 2 weeks of respiratory symptoms. Respiratory failure, started ECMO and dialysis. Elevated D-dimers found on 2 occasions. Within 6 weeks of the onset of symptoms, death due to ARDS and MOF	N/D	LUNGS: very heavy, significant consolidations in both upper and lower lobes HEART: cardiomegaly, 590 g OTHERS: signs of liver damage, intestinal mucositis and intestinal hemorrhage	LUNGS: ARDS in organizing phase, with extensive fibrinous exudates and diffuse thickening of the alveolar septa. Massive hyperplasia of the alveolar and bronchial epithelium, focal squamous metaplasia and typical concentric layered formations of loose connective tissue, with central aggregates of inflammatory cells	N/D	SARS CoV-2 RNA detected in lthe lungs	ARDS, MOF	Positive for SARS CoV-2 (pharyngeal)
Louis Maximilian Buja et al.	62	M	Obesity (BMI 33.8 Kg/m^2^)	N/D	N/D	N/D	Respiratory symptoms for a few days, found dead in his car	N/D	LUNGS: heavy HEART: 420 g. Mild coronary atherosclerosis. OTHERS: enlarged and congested spleen	LUNGS: Early stage DAD with multiple hyaline membranes, focal mild inflammation. CD68 + macrophages in the alveolar spaces. Reactive pneumocytes with cytomegaly, nucleomegaly, prominent nucleoli and mitotic figures. Squamous metaplasia. HEART: cardiomyocytes with moderately enlarged hyperchromatic nucleus and rare cardiomyocytes with degenerative vacuolar changes. CD3 + lymphocyte infiltrates in the epicardium LIVER: moderate macrovesicular steatosis KIDNEY: hyaline arteriolosclerosis and glomerulosclerosis. Viral particles in glomerular endothelial cells SPLEEN: enlarged. Red pulp expansion due to congestion and lymphoplasmacytic infiltrate. White pulp reduction. Scattered immunoblasts present near the edge of the white pulp and in the red pulp.	N/D	ELECTRON MICROSCOPY: neutrophils in the alveolar capillaries and fibrin in the alveolar spaces. No viral particles in the lungs or in the heart.Panel for hepatitis A, B and C: negative	Not specified	POST MORTEM positive for SARS CoV-2 (nasopharyngeal)
	34	M	Obesity (BMI 51.65 Kg/m^2^), HTN, heart failure with reduced EF (> 20%), DM2	Antibiotics, O_2_	Leukocytosis, mild anemia, mildly elevated troponin, elevated creatinine	Chest x-ray: cardiomegaly. Diffuse bilateral interstitial pulmonary opacities Chest CT: Diffuse circular ground glass opacities in the upper and lower lobes of both lungs. Dilated pulmonary artery, a sign of pulmonary hypertension. Cardiomegaly and traces of pericardial effusion	Hospital presentation: 4-day history of headache, shortness of breath, and productive cough with hemoptysis, 1-day of fever. Recurrent fever, hemoptysis and shortness of breath.Death 10 days after hospitalization due to respiratory failure and cardiac arrest	N/D	LUNGS: Extremely congested, with multiple hemorrhagic areas and multiple bilateral segmental thromboemboli HEART:1070 g, dilated hypertrophy. Mild coronary atherosclerosis	LUNGS: multiple acute segmental bilateral thromboemboli, infarcted areas and hemorrhage. Interstitial lymphocytic pneumonia. Microscopic thrombi found in some pulmonary arterioles. In the alveoli, multiple fibrin deposits not organized in hyaline membranesHEART: CD3+ epicardial lymphocytic infiltrates, myocardiocytic hypertrophy, multifocal interstitial fibrosis, scattered damaged cardiomyocytes. OTHERS: moderate hepatic steatosis. Thrombi in glomerular capillaries and in the peritesticular veins	N/D	Influenza virus and RSV test: negative	Not specified	Positive for SARS CoV-2 (nasopharyngeal)
	48	M	Obesity (BMI 35.2 Kg/m^2^)	N/D	N/D	N/D	The man was found dead in his residence	N/D	LUNGS AND AIRWAYS: 500 mL of purulent and opaque watery fluid collected in the right pleural cavity. Translucent yellowish material found focally in the visceral pleura, along the upper-middle interlobar fissure. Brownish-green exudate with fibrotic thickening in the parietal and visceral pleura of the lower lobe. Signs of empyema. Lungs were heavy. Tracheobronchial tree had hyperemic mucosa, without mucous plugs. HEART: 670 g. Mild coronary atherosclerosis. Both ventricles were dilated	LUNGS: in the right pleura, empyema. Atalectasis and DAD with hyaline membranes, fibrinous intra-alveolar exudate, abundant intracapillary megakaryocytes, intra alveolar macrophages and activated pneumocytes. Scattered neutrophils and intra alveolar hemorrhages HEART: multifocal lymphocytic infiltrates in the epicardium. Signs of acute damage. Thickening of coronary arteries with narrowing of the lumen. Hypertrophic cardiomyopathy OTHERS: moderate macrovesicular hepatic steatosis, portal lymphocytes, portal fibrosis and porto–portal bridging fibrosis. In the kidney, mild arteriolosclerosis with rare sclerotic glomeruli. In the spleen, lymphocyte depletion in the white pulp with absence of marginal areas, expanded red pulp with congestion and hemorrhage	N/D	Influenza virus test: negative	Not specified	POST MORTEM positive for SARS CoV-2 (nasopharyngeal)
Esther Youd et al.	88	F	Dementia	N/D	N/D	N/D	Died in a nursing home. No symptoms known	N/D	LUNGS AND AIRWAYS: Heavy lungs. Bilateral lobar pneumonia. Tracheal inflammation with presence of mucusHEART: minimal atheromatous plaques OTHERS: small, fibrotic kidneys. Brain atrophy	LUNGS: DAD with hyaline membranes, type 2 pneumocyte hyperplasia and enlargement of the alveolar walls and interstitium, with lymphocytic infiltrate	N/D	N/D	Not specified	POST MORTEM positive for SARS CoV-2 (trachea and lungs)
	86	M	HTN, COPD, heart failure, dementia	N/D	N/D	N/D	Died in a nursing home. Symptoms reported before death were: cough, fever, postural instability	N/D	LUNGS AND AIRWAYS: Heavy lungs, with signs of consolidation. Pulmonary edema, anthracosis. Tracheal inflammation with presence of mucus HEART: 592 g, minimal atheromatous plaques OTHERS: Chronic hepatic venous congestion. Fibrotic kidneys. Enlarged spleen with visible nodules	LUNGS: DAD with hyaline membranes, type 2 pneumocyte hyperplasia and enlargement of the alveolar walls and interstitium, with lymphocyte infiltrate. Bone marrow embolism HEART: chronic ischemic changes and contraction band necrosis SPLEEN: B-cell lymphoma undiagnosed in vivo	N/D	N/D	Not specified	POST MORTEM positive for SARS CoV-2 (trachea)
	73	F	Obesity, DM1, asthma, heart failure	N/D	N/D	N/D	Died at home. Reported shortness of breath before death	N/D	LUNGS AND AIRWAYS: Heavy lungs. Presence of consolidations, pulmonary edema and pleural adhesions. Tracheal inflammation with presence of mucus HEART: hypertrophic heart (582 g), focal coronary stenosis from atheroma and old myocardial fibrosis, no signs of myocarditis OTHERS: In the liver, chronic venous congestion. Fibrotic kidneys, with stones. Atheroma in the circle of Willis	LUNGS: DAD with hyaline membranes, type 2 pneumocyte hyperplasia and enlargement of the alveolar walls and interstitium, with lymphocyte infiltrate HEART: chronic ischemic changes	N/D	N/D	Not specified	POST MORTEM positive for SARS CoV-2 (trachea)
Lisa M. Barton et al.	77	M	Obesity (BMI 31.8 kg/m^2^), HTN, history of DVT, splenectomy, history of pancreatitis. Positivity to ANAs	N/D	N/D	N/D	Chills and intermittent fever without cough for 6 days. Weakness and shortness of breath. Cardiac arrest occurred during transport to the hospital	X-ray: bilateral pulmonary opacities	LUNGS: heavy, red/brown in color, edematous parenchyma and solid consistency. Right pleural adhesionsHEART: hypertensive cardiac damage with microscopic signs of acute ischemia, coronary heart disease KIDNEYS: arteriolosclerosis, oncocytoma OTHER: BPH	LUNGS: DAD in the acute phase, with hyaline membranes. Scattered chronic interstitial inflammation, consisting mainly of lymphocytes. Thrombi in small pulmonary arterial branches. Congestion of alveolar septal capillaries and focal edema in the air spaces. Mild chronic inflammation of the bronchi and bronchioles, with edema in the bronchial mucosa. Scattered CD3 + lymphocyte infiltrates in the alveolar septa, with rare CD20 + lymphocytes	N/D	Standard Panel for Respiratory Pathogens and swab for Influenza Virus: negative	COVID-19, with contributing factors such as coronary heart disease	POST MORTEM Positive for SARS CoV-2 (nasopharyngeal and lower airways)
	42	M	Obesity (31.3 kg/m^2^), history of myotonic dystrophy and intestinal obstructions	N/D	N/D	Chest CT: bilateral ground glass opacities, bilateral consolidations	Hospital presentation: critical condition with fever, cough and shortness of breath, abdominal pain. Death due to heart failure followed after a few hours	X-ray: bilateral pulmonary opacities	LUNGS: heavy, with both lower lobes dark red in color OTHER: Liver cirrhosis, gynecomastia, mild coronary atherosclerosis and testicular atrophy. Nephrosclerosis	LUNGS: foci of acute bronchopneumonia, with signs of aspiration pneumonia and foreign material. Neutrophils and histiocytes in the peribronchiolar air spaces. CD68 + in areas of bronchopneumonia. KIDNEY: tubular crystals	N/D	Standard panel for respiratory pathogens: negative	complications of liver cirrhosis with other significant factors (myotonic dystrophy, aspiration pneumonia and COVID-19)	POST MORTEM Positive for SARS CoV-2 (nasopharyngeal) Negative for SARS CoV-2 (lower airways)
Miroslav Sekulic et al.	81	M	Dementia, left lung mass, coronary artery disease, AF treated with biventricular pacemaker, congestive heart failure, PVD, DM, dyslipidemia, HTN, CKD, gout, smoker, cerebrovascular events, and UTI. Surgical history of carotid endarterectomy, left inguinal hernia repair and cataract surgery	O_2_	pancytopenia, elevated creatinine, moderately elevated urea and BNP	Chest x-ray: diffuse patchy opacities in the right lung and subtle patchy opacities in the lower lobe of the left lung.Chest CT: multifocal bilateral ground glass opacities, lung mass in the left lower lobe, thin left pleural effusion, moderate cardiomegaly, calcifications in the coronary arteries and in thoracic aorta	Hospital presentation: acute respiratory failure and fever. Cough, need for oxygen support until death, 5 days after hospitalization	N/D	LUNGS: heavy lungs. The parenchyma was congested and emphysematous. Mass in the lower left lobe HEART: 620 g, hypertrophy with signs of chronic ischemia, severe coronary stenosis, interstitial fibrosis	LUNGS: DAD in acute/exudative phase, with hyaline membranes, scattered squamous metaplasia of the distal airways and emphysematous changes. Minimal chronic submucosal inflammation in the bronchi and trachea. Large cell carcinoma, with metastases in the ipsilateral hilar and peribronchial lymph nodes KIDNEY: acute tubular damage in the context of chronic kidney damage	N/D	Blood cultures and urine cultures: negative. Test for Legionella, pneumococcus, HIV: negative. SARS CoV-2 RNA found in the lungs, bronchi and lymph nodes. In lower levels, also found in spleen, heart, liver, intestine and skeletal muscle	SARS CoV-2 infection in a setting of metastatic carcinoma, diabetes and ischemic cardiomyopathy, leading to respiratory failure	Positive for SARS CoV-2 (nasopharyngeal)
	54	M	HTN, DM2, overweight (BMI: 29.9 kg/m^2^)	Heparin, O_2_, remdesivir, vancomycin, piperacillin/tazobactam, propofol, vasopressors, intubation	Increased D-dimer, leukocytosis, lymphocytopenia, elevated creatinine, elevated liver enzymes	Chest x-rayn.1: diffuse bilateral opacities with areas of consolidation of the lower lobes Chest x-ray n.2: bilateral opacities with greater consolidation at the base of the right lung CHEST X-ray n.3: worsening of the pulmonary picture with greater interstitial engagement	Hospital presentation: 2-day history of shortness of breath and dry cough. Physical examination showed tachycardia and poor saturation (76%). Admitted to intensive care with acute hypoxemic respiratory failure, After performing positive blood cultures and urine cultures, antibiotic therapy was started. Intubated on day 10. Drop in blood pressure and heart rate. 12 days after the onset of symptoms, died of cardiac arrest	N/D	LUNGS: heavy and congested, with bilateral serohematic pleural effusion of 300 mL. Solid consistency HEART: 560 g, left ventricular hypertrophy and coronary atherosclerosis OTHERS: acute congestion of liver and spleen	LUNGS: DAD at various stages, with areas where prominent hyaline membranes and hyperplastic pneumocytes, intra-alveolar fibroblastic proliferation and interstitial edema. Intra-alveolar areas of acute inflammation Pulmonary edema, multinucleated giant cell clusters and foci of squamous metaplasia KIDNEY: signs of diabetic glomerulosclerosis and acute tubular necrosis	N/D	Influenza virus and RSV test: negative. Blood and urine cultures positive for coagulase negative Staphylococcus and Enterococcus faecalis. Viral RNA found in the lung parenchyma, bronchi, lymph nodes and spleen	SARS CoV-2 infection in a setting of diabetes and underlying cardiovascular problems, leading to respiratory failure and MOF	Positive for SARS CoV-2 (nasopharyngeal)
Chaofu Wang et al.	53	F	HTN, DM2	Arbidol, O_2_	severe lymphocytopenia, elevated IL-6 and CRP	Chest x-ray and Chest CT: unspecified severe lung lesions	Hospital presentation: 2-day history of cough, fever, and shortness of breath. ARDS and died of cardio-respiratory failure 8 days after admission	N/D	LUNGS: moderate bilateral pleural effusions and fibrotic pleural adhesions. Hepatization of lung tissue	LUNGS: Diffuse DAD. Alveolar spaces filled with macrophages, scattered lymphocytes and neutrophils. Massive serous and fibrinoid exudate in the alveolar spaces. Peribronchiolar metaplasia with interstitial fibrous hyperplasia. Occasional hyaline membranes, with thickened alveolar walls, proliferation of collagen fibers and lymphocytic infiltrates. Focal or patchy hemorrhages with fibrinous exudate. Thrombi in the small veins. Massive desquamation of the epithelium of bronchioles and alveoli. Proliferation and activation of type II pneumocytes, with inclusion bodies. HEART: multifocal myocardial degeneration, myocardial atrophy and interstitial fibrous hyperplasia. Scattered B (CD20 +) and T (CD3 +) lymphocytes KIDNEY: focal fibrotic glomeruli and edema of the tubular epithelium, with a slight infiltrate of B and T lymphocytes “	N/D	Immunohistochemistry for IL-6, IL-10, TNF-alpha: IL-6 and TNF-alpha expressed moderately in macrophages, whereas IL-10 massively expressed. Extensive and massive expression of PD-L1 by alveolar macrophages. Expression of ACE2 (SARS CoV-2 receptor) by hyperplastic type II pneumocytes and alveolar macrophages	Cardio-respiratory failure	Positive for SARS CoV-2 (nasopharyngeal)
	62	M	N/D	Peramivir, methylprednisolone, O_2_	severe lymphocytopenia, elevated IL-6 and CRP	Chest x-ray and Chest CT: unspecified severe lung lesions	Hospital presentation: 13-day history of cough, fever, and shortness of breath. ARDS and died of cardio-respiratory failure 10 days after admission	N/D	LUNGS: moderate bilateral pleural effusions and fibrotic pleural adhesions. Hepatization of lung tissue and consolidation	LUNGS: Diffuse DAD. Alveolar spaces filled with macrophages, with lymphocytes and neutrophils. Massive serous and fibrinoid exudate in the alveolar spaces. Abundant mucinous secretions in the bronchial tree. Peribronchiolar metaplasia with interstitial fibrous hyperplasia. Occasional hyaline membranes, with thickened alveolar walls, proliferation of collagen fibers. Focal or patchy hemorrhages with fibrinous exudate. The endothelial cells of the small pulmonary arteries were swollen. Presence of thrombi in the small veins. Massive desquamation of the epithelium of bronchioles and alveoli. Proliferation and activation of type II pneumocytes, with inclusion bodies. HEART: multifocal myocardial degeneration, myocardial atrophy and interstitial fibrous hyperplasia. Scattered B (CD20 +) and T (CD3 +) lymphocytes KIDNEY: focal fibrotic glomeruli and edema of the tubular epithelium, with a slight infiltrate of B and T lymphocytes	N/D	Immunohistochemistry for IL-6, IL-10, TNF-alpha: IL-6 and TNF-alpha moderately expressed in macrophages, IL-10 expressed massively. Extensive and massive expression of PD-L1 by alveolar macrophages. Expression of ACE2 (SARS CoV-2 receptor) by hyperplastic type II pneumocytes and alveolar macrophages	Cardio-respiratory failure	Positive for SARS CoV-2 (nasopharyngeal)
Zachary Grimes et al.	Middle age	M	HTN with anti-hypertensive therapy	Ceftriaxone, azithromycin, O_2_	Elevated ferritin, Elevated CRP	Chest x-ray: mild bipulmonary vascular congestion Chest x-ray n.2: new dense patchy opacities retrocardiac and in the middle of the left lung. Nebulous opacity in the lower right lobe	Hospital presentation: 9 days of fever, chills, myalgia, dry cough and dyspnea. Physical examination: temperature of 39.4 °C and a 92% SpO_2_. After starting antibiotic therapy for suspected bacterial superinfection, improvement in fever and leukocytosis. Oxygen support required. On day 9 after admission, weakness and worsening left chest pain and sudden cardiac arrest	N/D	LUNGS: pulmonary thromboembolism with right pulmonary artery occlusion. Multiple foci of solid lung parenchyma, compatible with pulmonary consolidations HEART AND VESSELS: cardiomegaly and left ventricular hypertrophy. DVT	LUNGS: alternating light pink and red areas (lines of Zahn), consistent with pulmonary thromboembolism	N/D	ELECTRON MICROSCOPY: Viral-like particles (60–120 nm) in the lung, located in cytoplasmic vacuoles in pneumocytes	Pulmonary thromboembolism	Positive for SARS CoV-2 (nasopharyngeal)
	Middle age	M	Asthma, HTN, pharmacologically controlled HIV infection	Broad spectrum antibiotics, O_2_, vasopressors, intubation	Elevated ferritin, Elevated CRP	Chest x-ray: multiple bilateral pulmonary opacities	Hospital presentation: fever, chills, productive cough and worsening dyspnea. On physical examination, a temperature of 38.4 °C and SpO_2_ 93%. Despite the use of broad spectrum antibiotics, CRP and Ferritin values continued to rise. Intubation and ventilatory support required. Hemodynamic instability and, after 8 days of hospitalization, death due to cardiac arrest	N/D	LUNGS: pulmonary thromboembolism with occlusion of the right and left pulmonary arteries. Multiple foci of solid lung parenchyma, compatible with pulmonary consolidations HEART AND VESSELS: cardiomegaly and left ventricular hypertrophy. DVT	LUNGS: alternating light pink and red areas (lines of Zahn), consistent with pulmonary thromboembolism	N/D	ELECTRON MICROSCOPY: Viral-like particles (60–120 nm) in the lung, located in cytoplasmic vacuoles in pneumocytes	Pulmonary thromboembolism	Positive for SARS CoV-2 (nasopharyngeal)
Kristine E. Konopka et al.	37	M	Asthma, DM2, in therapy with ipratropium bromide, albuterol, sitagliptin	HCQ, piperacillin/tazobactam, vancomycin, CS, ECMO, dialysis, intubation	N/D	Chest CT: multifocal ground glass opacities	Hospital presentation: 1-day history of fever, non-productive cough and myalgia. Worsening hypoxemia and, intubated, died after 9 days of hospitalization	N/D	LUNGS: heavy lungs, consolidation of the lung parenchyma AIRWAYS: mucous plugs in the conduction ways	AIRWAYS: paucicellular mucus plugs, goblet cell metaplasia, mucus gland hyperplasia and thickening of subepithelial basement membranes LUNG: DAD, hyaline membranes, interstitial edema and reactive pneumocytes. Rare fibrin thrombi in small vessels and in a small pulmonary muscular artery. Mild fibrinous exudate in distal air spaces without involvement of bronchi or bronchioles, with predominant inflammatory mononuclear cells and scattered neutrophils	N/D	N/D	ARDS due to SARS CoV-2	Positive for SARS CoV-2 (not specified)
Randall Craver et al.	17	M	N	N/D	N/D	N/D	Collapsed after 2 days of severe headache, dizziness, nausea and vomiting. It was later reported that he had complained of flu-like symptoms and a dry cough without fever, but both the PCR for Influenza A and B and the throat cultures performed at the time were negative	N/D	LUNGS: heavy and congested HEART: hypertrophic heart (500 g), soft and with mottled parenchyma. 80 mL of pericardial fluid in the cavity	LUNGS: congestion, focal acute hemorrhage and edema. Thickened bronchi membranes, mild chronic inflammation of the submucosa HEART: diffuse inflammatory infiltrates associated with multiple foci of myocyte necrosis. Minimal interstitial fibrosis LIVER: centrilobular congestion with minimal steatosis	negative	Tests for influenza A and B, parainfluenza and RSV: negative	Fulminant eosinophilic myocarditis	POST MORTEM positive for SARS CoV-2 (nasopharyngeal)
Lei Yan et al.	44	F	Obesity (BMI 41.5 Kg/m^2^)	O_2_, intubation, vasopressors, HCQ, azithromycin, tocilizumab	Lymphocytopenia, elevated CRP, elevated ESR, elevated Troponin I, elevated D-dimer, elevated IL-6	Chest x-ray: irregular bilateral peripheral opacities Echocardiography: severe hypokinesias with mild EF reduction	Hospital presentation: fever, cough and dyspnea for 1 week. Physical examination: tachypnea, tachycardia and desaturation. ARDS, MOF and diagnosis of reverse Tako Tsubo, with death after 6 days of hospitalization	N/D	LUNGS AND AIRWAYS: heavy lungs, pleuritis with flat, opaque spotted lesions. The pleura had large areas of intense erythema overlying regions of pulmonary consolidation. Mucous secretions in the bronchi, trachea and nostrils. The mucosal surface of the trachea and bronchi was edematous and erythematous. Enlarged parabronchial lymph nodes HEART: 410 g. Striations of the myocardial tissue of the right atrial wall, with thin myocardial trabeculae	LUNGS: severe edema and isolated areas of pulmonary infarction. Diffuse interstitial lymphocytic infiltrates and fibrinous exudate. DAD with hyaline membranes. Desquamation of pneumocytes with likely viral cytopathic effect. Non-necrotizing lymphocytic vasculitis in the pulmonary vessels. HEART: mild myxoid edema, mild myocardiocytic hypertrophy. Rare foci of CD45 + lymphocytes KIDNEY: focal evidence of acute tubular damage with flattened epithelial tubules and lumens containing desquamated epithelial cells, granular cast and Tamm-Horsfall protein. Congestion in peritiubular capillaries	N/D	ELECTRON MICROSCOPY: viral-like particles in altered pneumocytes (50–75 nm). Presence of fibrin microaggregates in the vessels and fibrinous exudates in the alveolar spaces. Enlarged interstitial fibroblasts and activated lymphocytes	ARDS, MOF, reverse Tako tsubo cardiomyopathy	Positive for SARS CoV-2 (nasopharyngeal)
J. Matthew Lacy et al.	58	F	DM2, obesity (BMI: 38 kg/m^2^), dyslipidemia, asthma, ulceration of the lower limbs	N/D	N/D	N/D	After 7 days of fever and respiratory distress, found dead at home during quarantine	N/D	LUNGS AND AIRWAYS: heavy and edematous. Areas of hemorrhage in the upper and middle right lobes and in the lower left lobe. Thick mucus in the airways. Enlarged hilar and mediastinal lymph nodes. HEART: 438 g, mild atherosclerosis in major coronary vessels and subrenal aorta KIDNEY: focal scars in the cortex CNS: hydrocephalus ex vacuo	LUNGS: widespread edema, presence of hyaline membranes. Mild septal mononuclear infiltrates, with hyperplasia of desquamating pneumocytes and focal multinucleated cells. Acute alveolar hemorrhages and foci of reactive alveolar foamy macrophages. Intra-alveolar fibrin deposits. HEART: hypertrophy of myocardiocytes with interstitial and perivascular fibrous tissue LIVER: mild steatosis and centrolobular congestion KIDNEY: arteriolosclerosis, mesangial sclerosis and hypercellularity, focal glomerulosclerosis. Incidental nodule in the adrenal cortex OTHERS: focus of papillary thyroid carcinoma	N/D	Influenza virus test: negative. Positive bacterial cultures for Staphylococcus aureus and Streptococcus viridans, interpreted as contamination or post-mortem artifacts	ARDS due to SARS CoV-2	POST MORTEM positive for SARS CoV-2 (lower airways)
Evan A. Farkash et al.	53	M	Obesity, dyslipidemia	HCQ, O_2_, furosemide, metolazone, intubation	Leukocytosis, reduced GFR	Chest x-ray: bilateral patchy opacities	Hospitalized for aortic dissection, which was surgically repaired. Re-intubation on day 6 due to hypoxemia. MOF and cardiac arrest, with death on postoperative day 12	N/D	LUNGS: widespread signs of DAD, edema and acute bronchopneumonia	LUNGS: DAD, hyaline membranes and edemaKIDNEY: mild autolysis	N/D	Respiratory Pathogen Standard Panel with swab for Influenza Virus: Negative. ELECTRON MICROSCOPY: abundant viral-like particles (65–91 nanometers) inside tubular epithelial cells, in areas of isometric vacuolation	MOF, AKI	Positive for SARS CoV-2 (not specified)
Diego Aguiar et al.	31	F	Obesity (BMI 61.2 kg/m^2^)	N/D	Elevated CRP	N/D	Found dead at home, in voluntary isolation after 7 days of cough. An opioid antitussive and ibuprofen found on the scene. Rectal temperature of 41.4 °C, 2 h after death	CT: diffuse bilateral ground glass opacities associated with panlobar consolidations and air bronchograms	LUNGS: Heavy, grossly solid and rubbery lungs, with bilateral hemorrhagic edema, pleural and tracheobronchial effusions. Heterogeneous areas of whitish consolidation OTHERS: skin petechiae, signs of shock	LUNGS: alveolar damage and edema, DAD in the exudative phase with the presence of hyaline membranes, fibrin deposits and moderate activated and desquamated pneumocytes. Alveolar exudate, moderate increase in intra-alveolar macrophages. Focal areas of intra alveolar hemorrhage and bacterial proliferation. Abundant septal and capillaries polymorphonuclear cells OTHER: Chronic tracheitis and microabscesses in the liver parenchyma	dextromethorphan found in the patient’s blood	Panel for influenza viruses A and B, RSV A and B, adenovirus, rhinoviruses, bocavirus, metapneumovirus, other coronaviruses and parainfluenza viruses 1–4: negative	Lung changes related to SARS CoV-2	POST MORTEM positive for SARS CoV-2 (lower airways)
Takuya Adachi et al.	84	F	N	Ampicillin/sulbactam, CS, lopinavir/ritonavir, morphine, O_2_	N/D	Chest x-ray: bilateral opacities Chest CT: ground glass opacities and consolidations, especially in the lower lobes	Fever, diarrhea and shortness of breath while on cruise, Admitted to the hospital with dyspnea and fever. ARDS and hypoxemia, died after 16 days	N/D	LUNGS AND AIRWAYS: Lungs partially brown in color, consolidated. Thickening of both pleurae HEART: dilation of the right ventricle GI: diffuse multiple punctate hemorrhages in the mucosa of the stomach and duodenum	LUNGS: signs of DAD, both in the exudative phase and in the organization. In the exudative tissues there were prominent hyaline membranes, in those in the organization phase desquamation, squamous metaplasia, hyaline membranes and inflammatory infiltrates with prominent plasma cells in the alveolar septa. Intra-alveolar hemorrhages, vascular congestion, type 2 pneumocyte hyperplasia. Also note syncytial multinucleated cells OTHER: Hemophagocytosis in the lungs, spleen and lymph nodes. The glomeruli of both kidneys were affected by microthrombi, suggesting a picture of DIC	N/D	Sputum culture: positive for Staphylococcus Aureus and Klebsiella Pneumoniae. SARS CoV-2 RNA found at low levels in blood and faeces, colon, liver and spleen. Not found in urine. Viral antigens found in alveolar epithelial cells in the first phase of DAD and in syncytial multinucleated cells	Respiratory failure due to SARS CoV-2	Positive for SARS CoV-2 (nasopharyngeal). POST MORTEM: Positive in both lower airways and upper airways, with muore copies of viral RNA in lower airways (bronchi)
Parisa Karami et al.	27	F	N	Azithromycin, ceftriaxone, oseltamivir, lopinavir/ritonavir, HCQ, meropenem, vancomycin, methylprednisolone, O_2_, intubation	Leukopenia, thrombocytopenia, elevated CRP, elevated LDH, elevated D-dimer	Chest x-ray: weak bilateral patchy opacities Chest CT: weak bilateral ground glass opacities and pleural thickening Chest CT n.2: pulmonary consolidations and pleural effusions	30 weeks pregnant, presented to the hospital with a 3-day history of respiratory symptoms, fever, cough and myalgia. Physical examination: tachypnea, fever, hypoxemia. No swab performed. MOF and death of patient and fetus. Post mortem diagnosis of COVID-19 by rt-PCR	N/D	N/D	LUNGS: focal hyaline membranes, pneumocytes proliferation and metaplastic changes. Cytopathic effects from viral infection. Lymphocytes and macrophages	N/D	N/D	MOF	Positive for SARS CoV-2 (not specified)
Christine Suess et al.	59	M	HTN, DM2	Antitussive	Lymphocytopenia	N/D	Presented to family physician with dry cough, fever and tachycardia, quarantined. Found dead at home 5 days later	CT: bilateral ground glass opacities and consolidations, slight pericardial and pleural effusions	LUNGS: heavy lungs, hemorrhages in the pleural surfaces. Pulmonary edema and diffusely solid and rubbery parenchyma. Bronchi fluid-filled. Dark red parenchyma, with scattered hemorrhagic areas. Enlargement of hilar lymph nodes	LUNGS: congestion and early stage DAD with hyaline membranes, proteinaceous exudate, alveolar hemorrhage and intra-alveolar fibrin deposition. Patchy distribution of intra-alveolar foamy macrophages in all lobes. Hyperplastic type II pneumocytes, with likely nucleolar cytopathic effects. In the epithelium of the bronchi, similar reactive picture. Increased number of intravascular megakaryocytes and slight patchy increase in interstitial lymphocytes. Some areas of bronchial metaplasia and fibrosis of the interstitium were present. In the lower lobes, focal neutrophilic infiltration in some air spaces and bronchial walls. Hyaline microthrombi were found in the pulmonary capillaries and some fresh thrombi in the pulmonary arterial branches. In the lymph nodes found several non-caseous granulomas LIVER: moderate macro and micro vesicular steatosis, with some necrotic hepatocytes around the central veins HEART: Patchy, non-specific pericardial infiltrations with inflammatory cell aggregates, including plasma cells and lymphocytes	N/D	Tests for influenza virus, RSV, rhinovirus, metapneumovirus, parainfluenza virus, adenovirus, enterovirus, bocavirus, other coronavirus: negative	ARDS from severe diffuse alveolar damage due to severe SARS CoV-2 infection	Positive for SARS CoV-2 (nasopharyngeal)
Monique Freire Santana et al.	71	M	HTN, DM2, CKD	O_2_, vasopressors, oseltamivir, HCQ, azithromycin, ceftriaxone, furosemide, heparin, intubation	Elevated urea, elevated creatinine and CRP, lymphocytopenia, neutrophilia	Chest x-ray: infiltrates and nodular consolidation in the lower right lobe	Transferred to a COVID-19 dedicated facility, required orotracheal intubation and pharmacological support. After 4 days of hospitalization, hemodynamic worsening, up to shock with irreversible hypotension, bradycardia and death	N/D	LUNGS: focal areas of consolidation in the right lower lobe	LUNGS: presence of well-defined Aspergillus structures, with hyphae and spores. Snoring pneumonia, fibrin thrombi occluding an artery, and squamous metaplasia were also present. Aspergillus was also in the pulmonary vessels	N/D	blood culture: negative for bacterial growth Detection of GM antigen in peripheral blood: positive PCR for aspergillus: positive	Shock	Positive for SARS CoV-2 (not specified)
James R. Stone et al.	76	F	Asthma, DM, HTN, dyslipidemia, osteoporosis, psoriasis. Therapy: atorvastatin, aspirin, hydrochlorothiazide, losartan, insulin, metformin, glipizide, citalopram, acetaminophen, cholecalciferol, folate, fluticasone, betamethasone, topical fluocinonide	Paracetamol, ceftriaxone, azithromycin, HCQ, metroprolol, furosemide, O_2_	hyperglycemia, elevated creatine kinase, elevated LDH, elevated Ferritin, elevated D-dimer	Chest x-ray: patchy opacities in the upper left lobe and near the hilum Chest CT with contrast: ground glass opacities and bilateral multifocal consolidations	Resident in a nursing home, presented with nasal congestion, confusion, incontinence and hypoxemia. Physical examination: lethargy, fever, hypertension and hypoxemia. On day 2, episodes of intermittent fever, worsening of delirium and hypoxemia. On day 3, onset of AF. On day 4, marked hyperpnea and hypoxemia despite ventilatory support. Palliative therapy started, died 36 h later	N/D	LUNGS: areas of pulmonary consolidation HEART: hypertrophic heart, dilation of the left ventricle. Mild coronary atherosclerosis	LUNGS: large area of alveolar parenchyma with preserved architecture, with thick hyaline membranes associated with focal desquamation of pneumocytes and congested capillaries. In some areas, the alveolar walls showed increased cellularity with some fibroblast-like spindle cells. These findings were consistent with an early exudative/proliferative phase of DAD. Rare foci of neutrophilic infiltrates and histiocytes in the alveolar spaces, suggesting a focal pneumonic process. Rare multinucleated giant cells AIRWAYS: presence of mucus and epithelial desquamation in the majority of the bronchi, with areas of squamous metaplasia. Few chronic perivascular inflammatory aggregates were also present. HEART: Diffuse CD68 + macrophage infiltrates in the myocardium, along with focal infiltrates of CD3 + T lymphocytes	N/D	Influenza A, B and RSV tests: negative. Immunohistochemistry for SARS CoV 1–2 (nucleocapsid protein) positive in alveolar macrophages and scattered pneumocytes	ARDS due to SARS CoV-2	Positive for SARS CoV-2 (nasopharyngeal)
Sociedad Espanola de Anatomìa Patologica	54	M	HTN, gout, migraine, OSAS, obesity (BMI: 30.9). Therapy with C-PAP	CS, lopinavir/ritonavir, HCQ, azithromycin, tocilizumab, enoxaparin, intubation, dialysis	Lymphocytopenia, elevated D-dimer, elevated LDH, elevated IL-6, elevated CRP, elevated ferritin	Chest x-ray: bilateral pulmonary opacities	Hospital presentation: 8-day history of chills, fever and cough. Steroid treatment, intubation. AKI and progressive desaturations, requiring tracheostomy and hemodialysis. Death after 25 days in ICU, due to pulmonary thromboembolism	N/D	LUNGS: heavy, hard and congested. Red and rubbery cut surface	LUNGS: reduced air spaces due to the thickening of the interstitial connective tissue. CD68 + macrophages. Hyperplastic pneumocytes with cytopathic effects. In the most affected areas of the lung, hyaline membranes compatible with DAD in the exudative stage. In 70% of the material lesions consistent with DAD in the proliferative phase and in the fibrotic phase. Intense septal thickening with an abundance of reactive fibroblasts. Abundance of thrombi in the middle and small pulmonary vessels. Some septal/alveolar calcium depositsKIDNEY: cortical necrosis and crystalluria (calcium oxalate)	N/D	Histochemical and immunohistochemical tests for other pathogens: negative	Pulmonary thromboembolism	Positive for SARS CoV-2 (nasopharyngeal)
Pedro Navarro Conde et al.	69	M	Low-grade non-invasive urothelial carcinoma of the bladder	Levofloxacin, ceftriaxone, O_2_	N/D	Chest x-ray: bilateral ground glass interstitial infiltrate in the lower lobes CT: negative for pulmonary thromboembolism	Hospital presentation: recent trip to China, fever, dyspnea, cough and non-acidotic hypoxia. Non-invasive ventilatory support, then, 2 days later, hospitalization in the ICU. Died 4 h after, due to shock and unresponsive AKI. Postmortem COVID-19 diagnosis	N/D	LUNGS: dark red in color, increased in weight and density. Pleural posterior adhesions HEART: mild stenosis of the aortic valve, slight thickening of the left ventricle and dilation of both ventricles OTHER: generalized congestion of other organs	LUNGS: edema and intra alveolar hemorrhage. DAD with desquamation of type II pneumocytes, and hyaline membranes, sometimes in the in the proliferative phase. Thrombi in the pulmonary vessels of medium caliber. Abundant intralveolar macrophages Cytopathic changes in pneumocytes and macrophages. Cells with large and hyperchromatic nuclei, similar to the smudge cells described in adenovirus pneumonia. Squamous metaplasia of pneumocytes. Inflammatory infiltrate with few lymphocytes and abundant macrophages. Emphysematous areas	N/D	Tests for influenza A and B, AH1N1, RSV, enterovirus, adenovirus, metapneumovirus, bocavirus, coronavirus 229, coronavirus NL63, coronavirus OC43, parainfluenza virus I, II, rhinovirus: negative. Immunohistochemical tests for Herpes Simplex, cytomegalovirus and EBV: negative	CLINICAL: severe bilateral CAP PATHOLOGICAL: SARS CoV-2 pneumonia	Positive for SARS CoV-2 (nasopharyngeal)
Fabian Heinrich et al.	59	M	Obesity (BMI: 32.8), HTN	N/D	N/D	N/D	Systemic symptoms during a cruise, with dyspnea. Hospitalized, developed fever and productive cough. Died after 6 days of hospitalization	CT: Moderate bilateral pleural effusions. Subpleural ground glass opacities and ground glass nodules in the center of the lungs. Multifocal reticular consolidations, especially in the central areas of both lungs	LUNGS: heavy and edematous lungs. Foamy hemorrhagic fluid in the upper respiratory tract. Areas of slightly nodular, dense and hyperemic plaura. Signs of acute hemorrhagic tracheitis and bronchitis, with patchy mucosal bleeding HEART: congestive cardiomyopathy, dilated atria and ventricles, extensive lipomatosis of the septum and cardiomegaly (600 g). Moderate arteriosclerosis	LUNGS: DAD, with prominent hyaline membranes, microvascular thromboembolism and activated pneumocytes, capillary congestion and protein-rich interstitial and intra-alveolar edema. Moderate mononuclear inflammatory infiltrate, mainly lymphocyte, absent granulocytes. Multinucleated syncytial cells in some alveoli. HEART: advanced interstitial and perivascular myocardial fibrosis, with biventricular lipomatosis. CNS: nonspecific immune response in the brainstem with perivascular and parenchymal CD8 + infiltrates. Minimal cerebral arteriosclerosis	N/D	Several SARS CoV-2 copies in the lungs	Respiratory failure due to SARS CoV-2	POST MORTEM Positive for SARS CoV-2 (nasopharyngeal and oropharyngeal)
Inga-Marie Schaefer et al.	66	F	SLE, RA, Pulmonary fibrosis, CKD, interstitial lung disease, MGUS, coronary heart disease, HTN	HCQ	N/D	Chest x-ray: bilateral opacities in the air spaces, especially in the periphery Chest CT: ground glass patchy opacities in the lower lobes and in the periphery	Hospital presentation: 2-week history of cough and fever. Died after 7 days	N/D	N/D	LUNGS: acute DAD, scattered foci in the organization phase and prominent hyaline membranes. Interstitial lung disease with bronchiectasis. Pulmonary thromboembolism. Diffuse interstitial and peribronchial lymphocytic inflammatory infiltrates, with intra-alveolar macrophages AIRWAYS: reactive squamous metaplasia. Minimal lymphocytic infiltrate in the edematous connective tissue of the airway walls	N/D	Post mortem immunohistochemistry for SARS CoV-2: positive in pneumocytes (<5 cells × 4 mm^2^)	SARS CoV-2 pneumonia resulting in respiratory failure	Positive for SARS CoV-2 (nasopharyngeal)
	57	M	HTN, DM, neurological impairment	N/D	N/D	Chest x-ray: Diffuse, bilateral opacities in the air spaces	Hospital presentation: cough, dyspnea and cardiac arrest. Died after 1 day	N/D	N/D	LUNGS: Acute DAD, with scattered foci in the organization phase and prominent hyaline membranes. Pulmonary thromboembolism. Diffuse interstitial and peribronchial lymphocytic inflammatory infiltrates, with intra-alveolar macrophages	N/D	Post mortem immunohistochemistry for SARS CoV-2: positive in pneumocytes (<5 cells × 4 mm^2^)	SARS CoV-2 pneumonia resulting in respiratory failure	Positive for SARS CoV-2 (nasopharyngeal)
	77	M	CKD, cardiomegaly, atherosclerosis, DM, dementia, limb amputation	N/D	N/D	Chest x-ray: diffuse bilateral opacities in the pulmonary air spaces and interstitium	Hospital presentation: cough, fever, dyspnea for 3 days and hypoxemic respiratory failure. Died after 3 days	N/D	N/D	LUNGS: Acute DAD, with scattered foci in the organization phase and prominent hyaline membranes. Pulmonary thromboembolism. Superimposed bacterial lobar pneumonia. Diffuse interstitial and peribronchial lymphocytic inflammatory infiltrates, with intra-alveolar macrophages AIRWAYS: reactive squamous metaplasia. Minimal lymphocytic infiltrate in the edematous connective tissue of the airway walls	N/D	post mortem immunohistochemistry for SARS CoV-2: positive in pneumocytes (<5 cells × 4 mm^2^)	SARS CoV-2 pneumonia resulting in respiratory failure	Positive for SARS CoV-2 (nasopharyngeal)
	50	M	Former smoker, UTI, aspergillus pneumonia, febrile neutropenia, relapsing B-cell acute lymphoblastic leukemia	N/D	N/D	Chest x-ray: Diffuse, bilateral opacities in the air spaces	Hospitalization for other reasons, developed cough, fever, hypoxemic respiratory failure and hematuria. Died after 11 days	N/D	N/D	LUNGS: acute DAD, with scattered foci in the organizing phase and prominent hyaline membranes. Superimposed bacterial lobar pneumonia. Aspergillus abscess. Prominent reactive hyperplasia of pneumocytes. Diffuse interstitial and peribronchial lymphocytic inflammatory infiltrates, with intra-alveolar macrophages AIRWAYS: reactive squamous metaplasia. Minimal lymphocytic infiltrate in the edematous connective tissue of the airway walls	N/D	post mortem immunohistochemistry for SARS CoV-2: positive in trachea (<5 cells × 4 mm^2^) and in pneumocytes (5–50 cells × 4 mm^2^)	SARS CoV-2 pneumonia resulting in respiratory failure	Positive for SARS CoV-2 (nasopharyngeal)
	68	F	Smoker, atherosclerosis, HTN, DM, COPD	N/D	N/D	Chest x-ray: diffuse bilateral opacities of the pulmonary air spaces and interstitium Chest CT: wide, subpleural sparing, bilateral ground glass opacities and consolidations	Hospital presentation: 7-day history of chest pain, fatigue and altered mental status. Died after 1 day	N/D	N/D	LUNGS: Acute DAD with prominent hyaline membranes. Prominent reactive hyperplasia of pneumocytes. Pulmonary thromboembolism. Diffuse interstitial and peribronchial lymphocytic inflammatory infiltrates, with intra-alveolar macrophages AIRWAYS: reactive squamous metaplasia. Minimal lymphocytic infiltrate in the edematous connective tissue of the airway walls	N/D	post mortem immunohistochemistry for SARS CoV-2: positive in trachea (5–50 cells × 4 mm^2^) and in lungs (pneumocytes and some macrophages) (> 50 cells × 4 mm^2^)	SARS CoV-2 pneumonia resulting in respiratory failure, complicating a recent myocardial infarction	Positive for SARS CoV-2 (nasopharyngeal)
	66	M	Former smoker, HTN, DM	HCQ	N/D	Chest x-ray: bilateral opacities in the air spaces, especially in the periphery	Hospital presentation: 7-day history of cough, fever and diarrhea. Died after 16 days	N/D	N/D	LUNGS: DAD and organizing lung injury. Diffuse interstitial and peribronchial lymphocytic inflammatory infiltrates, with intra-alveolar macrophages AIRWAYS: reactive squamous metaplasia. Minimal lymphocytic infiltrate in the edematous connective tissue of the airway walls	N/D	post mortem immunohistochemistry for SARS CoV-2: negative in lung and trachea	SARS CoV-2 pneumonia resulting in respiratory failure	Positive for SARS CoV-2 (nasopharyngeal)
	53	M	HTN, DM, CKD, NASH	Remdesivir	N/D	Chest x-ray: diffuse bilateral opacities in the pulmonary air spaces. Pneumomediastinum	Hospital presentation: 8-day history of cough, fever, and dyspnea. Died after 21 days	N/D	N/D	LUNGS: DAD and organizing lung injury. Superimposed bacterial lobar pneumonia. Pulmonary thromboembolism. Diffuse interstitial and peribronchial lymphocytic inflammatory infiltrates, with intra-alveolar macrophages AIRWAYS: reactive squamous metaplasia. Minimal lymphocytic infiltrate in the edematous connective tissue of the airway walls	N/D	post mortem immunohistochemistry for SARS CoV-2: negative in lung and trachea	SARS CoV-2 pneumonia resulting in respiratory failure	Positive for SARS CoV-2 (nasopharyngeal)
Mohammad Taghi Beigmohammadi et al.	58	M	HTN. Therapy: losartan, aspirin	HCQ, atazanavir, intubation	N/D	Chest CT: bilateral peripheral ground glass opacities, especially in the basal segments	Hospital presentation: fever, dyspnea, nausea and vomiting. Intubation, died 7 days after hospitalization	N/D	N/D	LUNGS: pulmonary edema, hyaline membranes, inflammation in the alveolar walls, hyperplasia of type II pneumocytes, alveolar macrophages, hemorrhagic areas, fibrinoid material in the walls of the vessels HEART: focal interstitial inflammation LIVER: mild portal inflammation, interface hepatitis, congestion, mild macro and microvescicular changes	N/D	N/D	Not specified	Positive for SARS CoV-2 (not specified)
	84	F	HTN. Therapy: amlodipine, aspirin, citalopram	HCQ, lopinavir/oseltamivir, intubation	N/D	Chest CT: bilateral peripheral ground glass opacities, especially along the basal segments	Hospital presentation: fever, dyspnea and myalgia. Intubation, died 3 days after hospitalization	N/D	N/D	LUNGS: pulmonary edema, hyaline membranes, fibrinous exudate, inflammation in the alveolar walls, alveolar macrophages, fibrinoid material in the vessel walls LIVER: minimal portal inflammation, severe congestion, mild macrovesicular and microvesicular steatosis, mild ballooning degeneration	N/D	N/D	Not specified	Positive for SARS CoV-2 (not specified)
	72	F	RA. Therapy: sulfasalazine, prednisolone, MTX	HCQ, levofloxacin, intubation	N/D	Chest CT: bilateral peripheral ground glass opacities, especially along the basal segments	Hospital presentation: fever, headache, nausea and vomiting. Intubated, died 15 days after hospitalization	N/D	N/D	LUNGS: pulmonary edema, fibrinous exudate, alveolar inflammation, type II pneumocyte hyperplasia, organization pattern and acute pneumonia, fibrinoid material in the vessel walls HEART: mild-moderate interstitial inflammation with LCA + and CD68 + cells LIVER: mild portal inflammation, mild interface hepatitis, mild fibrosis, moderate congestion, minimal macrovesicular steatosis, scattered biliary plugs	N/D	N/D	Not specified	Positive for SARS CoV-2 (not specified)
	72	M	HTN, DM with insulin treatment	HCQ, oseltamivir, atazanavir, levofloxacin, intubation	N/D	Chest CT: bilateral peripheral ground glass opacities, especially along the basal segments	Hospital presentation: fever, dyspnea and diarrhea. Intubation, died 4 days after hospitalization	N/D	N/D	LUNGS: pulmonary edema, hyaline membranes, inflammation in the alveolar walls LIVER: mild portal inflammation, mild congestion, minimal macrovesicular and microvescicular steatosis	N/D	N/D	Not specified	Positive for SARS CoV-2 (not specified)
	68	M	HTN, valvular regurgitation. Therapy: losartan, propranolol	HCQ, oseltamivir, intubation	N/D	Chest CT: bilateral peripheral ground glass opacities, especially along the basal segments	Hospital presentation: fever and dyspnea. Endocarditis and valve surgery. Development of respiratory symptoms, intubated, died after 19 days of hospitalization	N/D	N/D	LUNGS: pulmonary edema, hyaline membranes, fibrinous exudate, alveolar inflammation, type II pneumocyte hyperplasia, organization pattern, squamous metaplasia associated with bronchiolitis HEART: severe interstitial inflammation, myocardiocyte necrosis, LCA + and CD68 + cells, some CD3 + cells LIVER: mild portal inflammation and interface hepatitis, moderate congestion, mild ballooning degeneration, focal biliary plugs	N/D	N/D	Not specified	Positive for SARS CoV-2 (not specified)
	46	M	Peptic ulcer disease. Therapy: chlordiazepoxide, clidinium	HCQ, remdesivir, naproxen, cefepime, intubation	N/D	Chest CT: bilateral peripheral ground glass opacities, especially along the basal segments	Hospital presentation: fever, dyspnea, myalgia and pharyngodynia. Intubation, died after 16 days	N/D	N/D	LUNGS: pulmonary edema, fibrinous exudate, inflammation in alveolar spaces and walls, type II pneumocyte hyperplasia, organization pattern and acute pneumonia, hemorrhagic areas LIVER: minimal portal inflammation, moderate to severe congestion, mild macrovesicular and microvescicular steatosis, mild ballooning degeneration	N/D	N/D	Not specified	Positive for SARS CoV-2 (not specified)
	75	M	N	HCQ, oseltamivir, intubation	N/D	Chest CT: bilateral peripheral ground glass opacities, especially along the basal segments	Hospital presentation: fever, dyspnea and anorexia. Intubation, died after 6 days	N/D	N/D	LUNGS: pulmonary edema, hyaline membranes, fibrinous exudate, inflammation in alveolar spaces and walls, hyperplasia of type II pneumocytes, pattern of acute pneumonia with necrosis, fibrinoid material in vessel walls HEART: mild to moderate interstitial inflammation, LCA + and CD68 + cells, with some CD3 + cells LIVER: mild portal inflammation, mild interface hepatitis, confluent necrosis, moderate congestion, minimal-moderate steatosis, mild ballooning degeneration	N/D	N/D	Not specified	Positive for SARS CoV-2(not specified)
Sufang Tian et al.	78	F	Chronic lymphocytic leukemia	Antibiotics, antivirals, O_2_	elevated pro-BNP, elevated troponin, elevated LDH, leukocytosis	Chest CT 1: multiple bilateral ground glass opacities in the upper lobes, mostly on the rightChest CT 2: similar to the first CT, with thickening of the bronchi and vessels	Hospitalized for COVID-19 pneumonia at Wuhan University Zhongnan Hospital. Died after 22 days	N/D	N/D	LUNGS: DAD in acute phase with hyaline membranes, focal desquamation of pneumocytes and hyperplasia of type 2 pneumocytes, formation of syncytial giant cells. Focal lymphocytic infiltration LIVER: nuclear glycogenation of hepatocytes, mild focal macrovesicular steatosis, accumulation of neoplastic lymphocytes in the portal spaces HEART: mild focal edema, interstitial fibrosis and myocardial hypertrophy	N/D	RT-PCR for viral RNA in heart and liver samples: positive	SARS CoV-2 pneumonia	Positive for SARS CoV-2 (nasopharyngeal)
	74	M	Cirrhosis, variceal bleeding	Antibiotics, antivirals, O_2_	mildly elevated troponin, elevated LDH, leukocytosis, lymphocytopenia	Chest CT 1: patchy ground glass opacities, consolidations, air bronchogram Chest CT 2: additional consolidation in the left upper lobe	Hospitalized for COVID-19 pneumonia at Wuhan University Zhongnan Hospital. Died after 15 days	N/D	N/D	LUNGS: DAD in the acute phase, formation of hyaline membranes LIVER: pre-existing cirrhosis	N/D	RT-PCR for viral RNA in lung samples: positive	SARS CoV-2 pneumonia	Positive for SARS CoV-2 (nasopharyngeal)
	81	M	DM, HTN	Antibiotics, antivirals, O_2_	elevated troponin, elevated LDH, leukocytosis, lymphocytopenia	Chest x-ray 1: patchy opacities in both lungs, especially in the lower lobes Chest x-ray 2: worsening of the previous picture	Hospitalized for COVID-19 pneumonia at Wuhan University Zhongnan Hospital. Died after 23 days	N/D	N/D	LUNGS: Acute DAD with hyaline membranes, focal interstitial thickening, vascular congestion, mild interstitial inflammatory infiltrate LIVER: mild sinusoidal dilation, liver plaque necrosis, mild increase in sinusoidal lymphocytes	N/D	RT-PCR for viral RNA in liver samples: negative	SARS CoV-2 pneumonia	Positive for SARS CoV-2 (nasopharyngeal)
	59	M	Kidney transplant performed 3 months earlier	Antibiotics, antivirals, O_2_	elevated pro-BNP, elevated troponin, elevated LDH, mildly elevated GOT, elevated ALP, elevated gamma-GT, leukocytosis, lymphocytopenia	Chest CT 1: diffuse ground glass opacities, consolidation in the posterior segment Chest CT 2: additional visible air bronchogram	Hospitalized for COVID-19 pneumonia at Wuhan University Zhongnan Hospital. Died after 52 days	N/D	N/D	LUNGS: acute phase DAD with hyaline membranes, intra-alveolar hemorrhages, early organization, interstitial thickening, focal fibrinoid necrosis of small vessel walls. Evidence of consolidation consistent with bacterial superinfection LIVER: mild sinusoidal dilatation, hepatic plaque necrosis in the periportal and centrilobular area, scattered hyperplasia of Kuppfer cells, mild increase in sinusoidal lymphocytes, few lymphocytes in the portal tracts HEART: mild focal edema, interstitial fibrosis and myocardial hypertrophy	N/D	RT-PCR for viral RNA in heart and liver samples: negative	SARS CoV-2 pneumonia	Positive for SARS CoV-2 (nasopharyngeal)
Zsuzsanna Varga et al.	71	M	Kidney transplant, coronary heart disease, HTN	Intubation	N/D	N/D	Hospitalized with a diagnosis of COVID-19, mechanical ventilation. 8 days later died due to MOF	N/D	N/D	LUNGS: concentration of mononuclear cells, with congestion of many of the small pulmonary vessels OTHERS: Apoptotic bodies in the heart, small intestine and lung	N/D	ELECTRON MICROSCOPY: in the transplanted kidney, viral inclusions in endothelial cells	MOF	Positive for SARS CoV-2(not specified)
	58	F	DM, HTN, obesity	Dialysis	N/D	N/D	Progressive respiratory failure due to COVID-19, MOF, dialysis required. On day 16 of admission, mesenteric ischemia requiring surgery. STEMI infarction, circulatory failure and cardiac arrest	N/D	N/D	LUNGS: lymphocytic endothelitis KIDNEY: lymphocytic endothelitis LIVER: hepatocyte necrosis, lymphocytic endothelitis HEART: myocardial infarction, lymphocytic endothelitis GI: endothelitis of the submucosal vessels	N/D	N/D	Cardiac arrest	Positive for SARS CoV-2(not specified)
Zhe Xu et al.	50	M	N/D	O_2_, interferon alfa-2b, lopinavir, ritonavir, moxifloxacin, methylprednisolone	lymphocytopenia	Chest x-ray on admission: multiple bilateral patchy opacities Chest x-ray 2: progressive infiltrate and diffuse bilateral reticular opacities	Hospital presentation: fever, chills, cough, fatigue and shortness of breath, recent trip to Wuhan. On day 14 after the onset, hypoxemia and worsened dyspnea died due to cardiac arrest	N/D	N/D	LUNGS: Bilateral DAD with fibromyxoid cell exudate. Desquamation of the pneumocytes and hyaline membranes, suggestive of ARDS. Pulmonary edema, interstitial inflammatory infiltrates, dominated by lymphocytes. Multinuclear syncytial cells with enlarged atypical pneumocytes and viral cytopathic changes LIVER: moderate microvesicular steatosis and mild lobular and portal activity HEART: slight mononuclear interstitial infiltrate	N/D	Cytometric analyzes: lymphocytopenia, lymphocyte hyperactivation	Cardiac arrest	Positive for SARS CoV-2 (pharyngeal)
Christine M. Lovly et al.	56	M	DM2, smoker, COPD, small cell lung cancer. Therapy: doxycycline for possible pneumonia before diagnosis of carcinoma, then carboplatin, etoposide, atezolizumab	Methylprednisolone, infliximab, O_2_, vancomycin and piperacillin/tazobactam, vasopressors, intubation	High ferritin, high LDH	CT: spiculated mass in the lingula, 5 cm in diameter. Mediastinal lymphadenopathy and multiple liver masses CT 2: bilateral ground glass opacities with thickening of the interlobular septa, more pronounced in the right upper lobe CT 3: progression of bilateral ground glass opacities, reduction in the size of the previously detected tumor	1 month of left chest pain, dyspnea, cough, sinusitis, refractory to doxycycline therapy. Hospitalized, diagnosis of small cell lung cancer, treated and discharged. Returned for dyspnea and hypoxemia. Worsening conditions, intubation and mechanical ventilation, shock development, palliative care	N/D	N/D	LUNGS: widespread DAD, especially in the organizing phase, with greater involvement of the right lung. Thickened alveolar septae, alveolar foamy macrophages, desquamating epithelial cells, organizing fibromyxoid exudates, fibrin, hemorrhages and edema. Areas of fibrosis and fibrotic nodules in the alveolar spaces. Reactive epithelial cells. Immunohistochemistry revealed the presence of CD68 + cells in the interstitium, with some CD3 + and CD20 + cells	N/D	Serum antibodies to SARS CoV-2: positive for IgG and IgM (low positivity) IN SITU HYBRIDATION FOR VIRAL RNA: SARS CoV-2 RNA identified in the submucosal glands of the large airways, in the macrophages of a paratracheal lymph node and in the pulmonary interstitium. In the lungs, positive within the intra alveolar macrophages, in the alveolar walls and in desquamating cells	Shock	Positive for SARS CoV-2 (nasopharyngeal)

A: age; G: gender; Labor: Laboratory findings; Tox: Toxicological findings. af: atrial fibrillation; aki: acute kidney injury; alp: alkaline phosphatase; ami: acute myocardial infarction; anas: antinuclear antibodies; ards: acute respiratory distress syndrome; av: atrioventricular; bnp: brain natriuretic peptide; bph: benign prostatic hyperplasia; c-pap: continuous positive airway pressure; cap: community acquired pneumonia; ckd: chronic kidney disease; cns: central nervous system; copd: chronic obstructive pulmonary disease; crp: c-reactive protein; cs: corticosteroids; CT: Computed tomography; dad: diffuse alveolar damage; dic: disseminated intravascular coagulation; dm: diabetes mellitus; DM2: diabetes mellitus type 2; dvt: deep venous thrombosis; eber: epstein-barr encoding region; ebv: epstein barr virus; ecmo: extracorporeal membrane oxygenation; ef: ejection fraction; esr: erythrocyte sedimentation rate; gamma gt: gamma glutamyltransferase; gfr: glomerular filtration rate; gi: gastrointestinal; got: glutamic oxaloacetic transaminase; hcq: hydroxychloroquine; hhv: human herpesvirus; hlh: hemophagocytic lymphohistiocytosis; htn: hypertension; icu: intensive care unit; il interleukin; ldh: lactate dehydrogenase; mgus: monoclonal gammopathy of undetermined significance; mof: multi organ failure; mtx: methotrexate; n/d: not described; n: none; nash: nonalcoholic steatohepatitis; net: neuroendocrine tumor; osas: obstructive sleep apnea syndrome; osc: other significant conditions; peg: percutaneous endoscopic gastrostomy; pvd: peripheral vascular disease; ra: rheumatoid arthritis; rsv: respiratory syncytial virus; rt-pcr: real time polymerase chain reaction; sle: systemic lupus erythematosus; uc: ulcerative colitis; uti: urinary tract infection.

**Table 2 diagnostics-11-00190-t002:** Type of post-mortem examination, role of SARS CoV-2, comparison between scores and inter-raters agreement.

Author	Age	Gender	Type of Examination	Role of SARS CoV-2	CSS R1	CSS R2	CSS R3	H Score
**Benjamin T Bradley et al.**	57	M	Complete autopsy	Cause of death	3	3	3	-
	74	F	Partial autopsy (no cranial cavity)	Contributing factor	2	2	2	-
	54	M	Partial autopsy (no cranial cavity)	Contributing factor	2	3	3	-
	74	M	Partial autopsy (no cranial cavity)	Contributing factor	3	3	3	-
	73	F	Partial autopsy (no cranial cavity)	Contributing factor	3	3	3	-
	84	F	Partial autopsy (no cranial cavity)	Contributing factor	3	3	3	-
	71	M	Partial autopsy (no cranial cavity)	Contributing factor	2	3	2	-
	76	F	Complete autopsy	Contributing factor	2	2	2	-
	75	F	Partial autopsy (no cranial cavity)	Contributing factor	3	3	3	-
	84	M	Complete autopsy	Significant factor	2	2	2	-
	81	F	Complete autopsy	Contributing factor	3	3	3	-
	42	F	Complete autopsy	Contributing factor	3	3	3	-
	71	M	Complete autopsy	contributing factor	1	1	1	-
	73	F	Partial autopsy (no cranial cavity)	Contributing factor	3	3	3	-
**Dominic Wichmann et al.**	52	M	Complete autopsy	Cause of death	3	3	3	2
	70	M	Complete autopsy	Cause of death	3	3	3	3
	71	M	Complete autopsy	Cause of death	3	3	3	2
	63	M	Complete autopsy	Cause of death	3	3	3	2
	66	M	Complete autopsy	Cause of death	2	2	2	1
	54	F	Complete autopsy	Cause of death	2	1	1	1
	75	F	Complete autopsy	Cause of death	3	3	3	1
	82	M	Complete autopsy	Cause of death	2	2	2	1
	87	F	Complete autopsy	Cause of death	1	2	2	3
	84	M	Complete autopsy	Cause of death	2	2	2	2
	85	M	Complete autopsy	Cause of death	3	3	3	1
	76	M	Complete autopsy	Cause of death	3	3	3	2
**Andrey Prilutskiy et al.**	72	M	Complete autopsy	Not specified	2	3	3	-
	91	M	Complete autopsy	Not specified	2	3	3	-
	72	M	Complete autopsy	Not specified	3	3	3	-
	64	F	Complete autopsy	Not specified	3	3	3	-
**Hans Bösmüller et al.**	78	F	Partial autopsy (no cranial cavity)	Not specified	2	2	3	-
	78	M	Partial autopsy (no cranial cavity)	Not specified	3	3	3	-
	72	M	Partial autopsy (no cranial cavity)	Not specified	2	2	2	-
	59	M	Partial autopsy (no cranial cavity)	Not specified	3	3	3	-
**Louis Maximilian Buja et al.**	62	M	Partial autopsy (no cranial cavity)	Not specified	3	3	3	-
	34	M	Partial autopsy (no cranial cavity)	Not specified	2	2	2	-
	48	M	Complete autopsy	Not specified	3	3	3	-
**Esther Youd et al.**	88	F	Complete autopsy	Not specified	3	3	3	-
	86	M	Complete autopsy	Not specified	3	2	2	-
	73	F	Complete autopsy	Not specified	3	3	3	-
**Lisa M. Barton et al.**	77	M	Complete autopsy	Cause of death	3	3	3	-
	42	M	Complete autopsy	Significant factor	1	1	1	-
**Miroslav Sekulic et al.**	81	M	Partial autopsy (no cranial cavity)	Cause of death	2	2	2	-
	54	M	Partial autopsy (no cranial cavity)	Cause of death	2	2	2	-
**Chaofu Wang et al.**	53	F	Complete autopsy	Not specified	3	3	3	-
	62	M	Complete autopsy	Not specified	3	3	3	-
**Zachary Grimes et al.**	Middle age	M	Complete autopsy	Not specified	3	3	3	-
	Middle age	M	Complete autopsy	Not specified	3	3	3	-
**Kristine E. Konopka et al.**	37	M	Complete autopsy	Cause of death	3	3	3	-
**Randall Craver et al.**	17	M	Complete autopsy	Not specified	U	U	0	-
**Lei Yan et al.**	44	F	Partial autopsy (no cranial cavity, internal organs left in situ)	Not specified	2	3	3	-
**J. Matthew Lacy et al.**	58	F	Complete autopsy	Cause of death	3	3	3	-
**Evan A. Farkash et al.**	53	M	Partial autopsy (no cranial cavity)	Cause of death	1	2	2	-
**Diego Aguiar et al.**	31	F	Complete autopsy	Cause of death	3	3	3	-
**Takuya Adachi et al.**	84	F	Partial autopsy (no cranial cavity)	Cause of death	2	2	2	-
**Parisa Karami et al.**	27	F	Partial autopsy (only lungs reported)	Not specified	3	3	3	-
**Christine Suess et al.**	59	M	Complete autopsy	Cause of death	3	3	3	-
**Monique Freire Santana et al.**	71	M	Complete autopsy	Not specified	1	1	1	-
**James R. Stone et al.**	76	F	Partial autopsy (only heart and lungs examined macroscopically)	Cause of death	3	3	3	-
**Sociedad Espanola** **de Anatomìa Patologica**	54	M	Partial autopsy (no cranial cavity, internal organs left in situ)	Not specified	3	3	3	-
**Pedro Navarro Conde et al.**	69	M	Partial autopsy (no cranial cavity)	Cause of death	3	3	3	-
**Fabian Heinrich et al.**	59	M	Complete autopsy	Cause of death	3	3	3	1
**Inga-Marie Schaefer et al.**	66	F	Post mortem histological samples (lung, airways)	Cause of death	3	3	3	-
	57	M	Post mortem histological samples (lung, airways)	Cause of death	3	3	3	-
	77	M	Post mortem histological samples (lung, airways)	Cause of death	2	2	2	-
	50	M	post mortem histological samples (lung, airways)	Cause of death	2	2	2	-
	68	F	Post mortem histological samples (lung, airways)	Cause of death	3	3	3	-
	66	M	Post mortem histological samples (lung, airways)	Cause of death	3	3	3	-
	53	M	Post mortem histological samples (lung, airways)	Cause of death	2	2	2	-
**Mohammad Taghi Beigmohammadi** **et al.**	58	M	Post mortem histological samples (lung, airways)	Not specified	3	3	3	-
	84	F	Post mortem histological samples (lung, airways)	Not specified	3	3	3	-
	72	F	Post mortem histological samples (lung, airways)	Not specified	3	3	3	-
	72	M	Post mortem histological samples (lung, airways)	Not specified	3	3	3	-
	68	M	Post mortem histological samples (lung, airways)	Not specified	2	3	3	-
	46	M	Post mortem histological samples (lung, airways)	Not specified	3	3	3	-
	75	M	Post mortem histological samples (lung, airways)	Not specified	3	3	3	-
**Sufang Tian et al.**	78	F	Post mortem histological samples (lung, airways)	Cause of death	3	3	3	-
	74	M	Post mortem histological samples (lung, airways)	Cause of death	3	3	3	-
	81	M	Post mortem histological samples (lung, airways)	Cause of death	3	3	3	-
	59	M	Post mortem histological samples (lung, airways)	Cause of death	2	3	3	-
**Zsuzsanna Varga et al.**	71	M	Post mortem histological samples (lung, airways)	Not specified	2	2	2	-
	58	F	Post mortem histological samples (lung, airways)	Not specified	2	2	2	-
**Zhe Xu et al.**	50	M	Post mortem histological samples (lung, airways)	Cause of death	3	3	3	-
**Christine M. Lovly, M.D. et al.**	56	M	Post mortem histological samples (lung, airways)	Not specified	1	3	3	-

CSS: COVID-19 Significance Score. F: female. H: Hamburg. M: male. R: rater.

## Data Availability

Data are available from the author upon request.

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
