# Peer review of "Died with or Died of? Development and Testing of a SARS CoV-2 Significance Score to Assess the Role of COVID-19 in the Deaths of Affected Patients"

_diagnostics, 2021, doi:10.3390/diagnostics11020190_

Round 1

Reviewer 1 Report

Authors submitted a review of literature concerning deaths involving Covid-19, witch were over 1.5 million recorded worldwide at the moment of writing the article.

Giorgetti et al. evaluate the application of the COVID-19 Significance Score – CSS - in the classification of SARS CoV-2-related fatalities, comparing it with Hamburg Score, another rating scale currently available.

In the “Abstract” paragraph, the Authors wrote “other rating scales currently available” however, reading the article I found just the Hamburg Scale; I suggest to correct the period informing about the comparison with the only Scale really used.

Introduction and CSS sections

The “Introduction” paragraph summarizes the events affected the whole world starting in December 2019 from the urban area of Whuan, concerning the spreading of the new form of corona-virus SARS CoV-2.

The Authors continue their review describing the CSS classification category of fatalities involving Covid-19, well explaining the prevalent features must be taken into consideration, including post-mortem data, circumstances of death, post-mortem imaging, macroscopic and microscopic autopsy findings and toxicological evaluation.

Material and methods

Data was collected from May to August 2020, searching on Pubmed database and an Excel document organized in two sections concerning “In vivo” data and “Post-mortem” data was created.

The 3.2. paragraph offers a guiding tool to facilitate the application of CSS, considering both “in vivo” and “post-mortem” data to collect.

The next section (3.3) focused attention on the alternative Hamburg score: the Scale propose a classification system for death involving Covid-19, reporting post-mortem data collected from the first 80 autopsies carried out in Hamburg on patients positive to COVID-19.

Both scale rank Covid-19 related death into categories, from 0 (Covid-19 is merely an occasion) to 3 (Covid-19 is the leading cause of death) according to CSS and from 1 (definite Covid-19 death) to 4 ( SARS-CoV-2 detection with cause of death not associated to Covid-19) according to Hamburg Scale.

Results

After opportune works exclusion (data from the first 80 autopsies carried out in Hamburg were excluded), the Authors examined 30 articles, corresponding to 84 autopsy; results are reported in the Table 1.

Analysed items are the following: Comorbidities, Past D, Therapy, Labor, Imaging, COD, Imaging, Macroscopic features, Microscopic features, Tox, Additional analyses, Cause of death, Swabs.

Analysis of results is complete and well explained, even in graphical illustrations.

The Authors proceeded to CSS (comparing the score assigned by three independent blinded investigators) and HS application to each case: results are shown in Table 2 and well exposed.

About Table 1, I suggest to better explain the column named “Past D” and “COD”, evaluating if related information can be omitted to simplify.

Discussion and Conclusion

The section is well articulate; the Authors performed an accurate review of literature concerning the diagnostic tests developed, the epidemiology of the victims and the rate of comorbidity, the history of the disease, reported symptoms, laboratory alterations and macro as well as microscopic findings of the cases collected (resulted in line with those reported by other works), performed imaging analysis, post-mortem examination.

Forensic approach appeared rather worrying because 55 post-mortem examinations were not complete; indeed, Giorgetti et al. concluded that the exclusion of some organs (mostly often, the brain) or the loss of a global view on the health status of the victim might lead to false conclusions.

Focusing the aim of the article, observing the CSS applied to the collected cases, it can be noted that most of the deceased fall into the category "deaths from COVID-19”, similar results is reported in other studies; the agreement found by three blinded and independent raters allows to hypothesize that the CSS is an easy tool which could be applied in the everyday routine of post-mortem examination on SARS-CoV-2 positive deceased. 

The 80 cases described by Edler et al. do not contain extractable information, particularly regarding post-mortem findings; therefore, they were not included in the database of the review.

The Authors, in fact, believe that a more comprehensive overview, as well as a valorisation of past history and of the status of the other organs and functions are needed, underling the importance of collecting in vivo data when performing a post-mortem assessment.

Conclusions offered are totally supported by data examined, deposing for CSS to be a useful tool helping in the assessment of the cases of deaths involving Covid-19 to reduce the inhomogeneities in forensic evaluation of SARS CoV-2 related deaths.

REVIEWER CONSIDERATIONS

The article propose an analytic approach to determinate the cause of death in patients with in vivo or post-mortem positivity of the swab for SARS CoV-2, to discriminate if “Died with or died of” Covid-19.

The manuscript concerns an important problem for the Forensic Pathologist called to discriminate the cause of death involving Covid-19; the article represents an original and usefull tool to assist during the whole post-mortem examination.

The Authors applied the CSS to 84 post-mortem examinations founded in literature, considering the 80 cases studied by Edler et al., not usable because they do not contain extractable information regarding post-mortem findings.

Used methods are outlined and reproducible; the results are clearly explained together with the presentation in Table and Figures.

The Authors conclude suggesting that, when the cause of death is difficult to be ascertained, a high degree of suspicion for COVID-19 should be maintained, and this probably had a reflection in the high degree of CSS 3 and 2 assigned.

Conclusions appear justified by the results showed.

About the References section, the list does not contain errors and all important references are mentioned.

I consider the review well conducted by Authors and conclusions are shareable to guide a forensic approach to establish the cause of death involving Covid-19.

To finish, the submitted review “Died with or died of? Development and testing of a SARS CoV-2 significance score to assess the role of COVID-19 in the deaths of affected patients” can be Accepted after Minor Revisions, as previously explained in the text:

-) In the “Abstract” paragraph, the Authors wrote “other rating scales currently available” however, reading the article I found just the Hamburg Scale; I suggest to correct the period informing about the comparison with the only Scale really used;

-) About Table 1, I suggest to better explain the column named “Past D” and “COD”, evaluating if related information can be omitted to simplify.

Reviewer 2 Report

This is an excellent and well-timed paper addressing a core question on Covid-19 mortality. It is well researched. Its limitations are well set out by the authors but perhaps not strongly enough on the issue of the small number of full and partial autopsies reported out of the total worldwide mortality from Covid-19 (including direct, contributory and late contributory categories of deaths). It would have been a helpful adjunct if the authors had alluded to part of the fuller reasons why so few autopsies are performed on Covid-19 decedents and any evidence about SARS-CoV-2 virus survival, its transmissibility and infectivity post mortem. There is no reference to the principles of death certification and registration from the legal framework point of view which clinicians and deaths investigators (including Coroners) must observe and how this interacts with the WHO clinical definitions and ICD classification of Covid-19 deaths. The literature review is excellent but also highlights the limited post mortem studies and numbers. Table 1 is a superb detail but very long and may paradoxically detract from the publication. Does the excellent Table 2 not detail adequately the data needed for this paper? The clinical summary is excellent. Perhaps the median age of the decedents would add to the analysis. The application of the CSS and its comparison with the Hamburg score is very good. However, Covid-19 related deaths diagnoses remain primarily a CLINICAL diagnosis and a brief cross reference to studies on the QCOVID prediction algorithm would be beneficial. The contention that the CSS could stand alone without reference to e.g QCOVID or other algorithms arising from the clinical diagnoses and diagnostic tools is a weakness of this paper in light of the small number of full autopsies reported in the literature to date. On a small comment, the harmonised international post mortem protocol of reference 79 has been updated in 2014 by the European Council of Legal Medicine. Overall an excellent study and review but subject to these comments.

Reviewer 3 Report

In this article the Authors present an overview of the literature cases of deaths involving COVID-19. The article assesses the application of the COVID-19 Significance Score in the classification of SARS CoV-2-related fatalities, comparing it with other rating scales currently available. According to the Authors the COVID-19 Significance Score used after a complete accurate post-mortem examination,coupled to the retrieval of in vivo data, post-mortem radiology, histology and toxicology, as well as to additional required analyses (e.g. electronic microscopy) is a useful, and concise tool in the assessment of the cause of death and the role played by this virus.

The topic of the article is very interesting, and it has a fairly good impact in scientific research on COVID-19.

English is quite good. Abstract is well written and focuses on the topic.

Introduction: the introduction is well written and offers a broad overview of COVID-19 and the classification of related COVID-19 deaths. Literature research is good, but it is not complete. In my opinion, the authors should add more details about the Forensic articles on COVID-19. For example, articles discussing this topic should be cited: “Barranco R, Ventura F. The role of forensic pathologists in coronavirus disease 2019 infection: The importance of an interdisciplinary research. Med Sci Law. 2020 Jul;60(3):237-238. doi: 10.1177/0025802420927825. Epub 2020 May 21. PMID: 32437227; Firth J. Covid-19 current advice for pathologists. Pathologica. 2020 Jun;112(2):55-56. doi: 10.32074/1591-951X-12-20. Epub 2020 Mar 17. PMID: 32292181”.

Materials and Methods: the methodology of the study appears correct as well. The authors describe the aims and the criteria of the study. The authors designate a complete and very detailed post-mortem investigation which included circumstances of death, macroscopic and microscopic examination, toxicological and radiological examinations. The description of the literature search needs to be expanded. Authors must specify in detail the inclusion criteria, the exclusion criteria and the search method.

Results: the results are clear. Tables are sufficiently exhaustive for the authors' purpose. In the text, the authors should insert two paragraphs reporting the main macroscopic and histological findings.

Discussion: the Authors should cited other articles about the postmortem nasopharyngeal swabs. For example, the article: “Ventura F, Barranco R. Cadaveric Nasopharyngeal Swab in Coronavirus Disease 2019 Infections: Can it be Useful for Medico-Legal Purposes? Am J Forensic Med Pathol. 2020 Sep;41(3):238-239. doi: 10.1097/PAF.0000000000000560. PMID: 32804452” should be cited.

For the rest, the Authors' message is useful. Surely in doubtful cases where we do not understand whether the patient died "from" or "with" COVID-19, the autopsy should be performed. The creation of a register that contains all the autopsies performed on patients affected by COVID-19 could be helpful.

In summary, after a minor revision, this paper could reach the requested level to be publishable in Diagnostics.

Round 2

Reviewer 2 Report

The authors are to be thanked for their prompt and comprehensive reply and attention to the comments, observations and recommendations. They have addressed all the points raised.